



# 1 Hydrogeological controls on the spatio-temporal variability of surge-
# 2 induced hydraulic gradients along coastlines: implications for beach
# 3 surface stability

Anner Paldor[1], Nina Stark[2], Matthew Florence[2], Britt Raubenheimer[3], Steve Elgar[3], Rachel Housego[3,4],
Ryan S. Frederiks[1], Holly A. Michael[1,5]
[1] Department of Earth Sciences, University of Delaware, Newark, DE, USA.
[2] Department of Civil and Environmental Engineering, Virginia Polytechnic Institute and State University. Blacksburg, VA,
USA.
[3] Department of Applied Ocean Physics and Engineering, Woods Hole Oceanographic Institution, Woods Hole, MA, USA.
[4] MIT-WHOI Joint Program in Oceanography, 86 Water St., Woods Hole, MA, USA.
[5] Department of Civil and Environmental Engineering, University of Delaware, Newark, DE, USA.
*Correspondence to*: Holly A. Michael (hmichael@udel.edu)
**Abstract.** Ocean surges pose a global threat for coastal stability. These hazardous events alter flow conditions and pore
pressures in flooded beach areas during both inundation and subsequent retreat stages, which can mobilize beach material,
potentially enhancing erosion significantly. In this study, the evolution of surge-induced pore-pressure gradients is studied
through numerical hydrologic simulations of storm surges. The spatiotemporal variability of critically high gradients is
analyzed in 3D. The analysis is based on a threshold value obtained for momentary liquefaction of beach materials under
groundwater seepage. Simulations of surge events show that during the run-up stage, head gradients can rise to the calculated
critical level landward of the advancing inundation line. During the receding stage, critical gradients were simulated seaward
of the retreating inundation line. These gradients reach maximum magnitudes just as sea level returns to pre-surge level, and
are most accentuated beneath the still-water shoreline, where the model surface changes slope. The gradients vary along the
shore owing to variable beach morphology, with the largest gradients seaward of intermediate-scale (1-3m elevation)
topographic elements (dunes) in the flood zone. These findings suggest that the common practices in monitoring and mitigating
surge-induced failures and erosion, which typically focus on the flattest areas of beaches, might need to be revised.



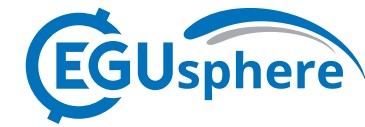

## 1 Introduction

Groundwater seepage can destabilize land areas, especFially at the interface between terrestrial and submerged systems (Iverson, 1995; Iverson & Major, 1986; Iverson & Reid, 1992; Schorghofer et al., 2004; Stegmann et al., 2011). Recent studies have examined the characteristics of pore pressure behavior, the associated groundwater seepage, and its effect on the stability of geomaterials (soils, rocks, etc.), including field observations (Mory et al., 2007; Sous et al., 2016), physical experiments (Schorghofer et al., 2004; Sous et al., 2013), numerical simulations (Orange et al., 1994; Rozhko et al., 2007; Schorghofer et al., 2004), and analytical models (Sakai et al., 1992; Yeh & Mason, 2014). There are several examples of seepage-induced failure of the surface (i.e. the mobilization of the soil skeleton) from around the world, including Japan (Yeh & Mason, 2014), California (Orange et al., 2002), and France (Sous et al., 2016; Stegmann et al., 2011).

Soil liquefaction occurs when pore pressures in the geomaterial rise to a point where its effective stress drops to zero and the material is fluidized, and thus acts as a liquid. At the coast, ocean (waves, surge, tides, inundation) and terrestrial (groundwater heads, precipitation, and overland flows) processes concurrently contribute to changing pore pressures in beach and nearshore sediments, and changes in pore pressure distributions and gradients could induce failure of the surface. Ocean effects on pore pressures, groundwater flow, and seepage occur due to wind waves, storm surges, and tsunamis. For example, a 1D analytical model suggests that during a tsunami, vertical hydraulic gradients can destabilize sediments and increase the potential for sediment momentary liquefaction, consistent with laboratory experiments (Abdollahi & Mason, 2020; Yeh & Mason, 2014). Laboratory experiments (Sous et al., 2013) suggest that the magnitude of hydraulic gradients in the beach due to infiltration from sea-swell and infragravity waves depend on the wave frequency, cross-shore position, water table overheight, and the presence of standing waves. A large-scale (250 m) flume study of a barrier island showed that waves can alter the coastal groundwater head distribution significantly, and can change cross-island and local (under the ocean beach) hydraulic gradient directions (Turner et al., 2016). Field observations of pore-pressures over several tidal cycles in a microtidal beach (Sous et al., 2016) suggest that breaking-wave-driven onshore increases in the water surface (setup) over the 10 m nearest the shoreline induced groundwater head changes of O(0.1 m) (Sous et al., 2016). Furthermore, density-driven flow at the subsurface transition zone between fresh terrestrial groundwater and saline groundwater can produce intense, localized seepage (Burnett et al., 2006). Rapid changes in seepage characteristics (locations, magnitudes, direction) during extreme events may lead to sediment liquefaction (i.e., loss of particle-to-particle contacts and sediment effective stresses) and mobilization, resulting in erosion and structure destabilization.

Observations, theories, and simulations have shown that the pore-pressure changes owing to energetic ocean waves can reduce effective stresses and may cause liquefaction (Chini & Stansby, 2012; Mory et al., 2007; Sakai et al., 1992; Sous et al., 2013; Yeh & Mason, 2014). Measured pore-pressure changes in beach sediments during intense waves suggest that momentary liquefaction may occur at shallow depths (<1 m) below the surface (Mory et al., 2007), consistent with theory (Sakai et al., 1992). Analytical solutions for the effective stress in an idealized seabed suggest that waves can alter the stresses in the upper meters of the seafloor significantly (Mei & Foda, 1981; Sakai et al., 1992). Simulations of a theoretical 2D porous medium,





where an increase in pore pressure is applied at the bottom of the layer from a point source, revealed that different spatial
failure patterns (i.e. the geometry of the slip surface) can occur under various stress regimes (i.e. distribution of stresses in the
soil) (Rozhko et al., 2007), although the process that leads to the simulated change in the pore-pressure distribution was
unexplored.
Apart from waves, storm surges also could alter the onshore hydrogeological regime and potentially reduce the stability of the
beach surface, yet surges have not been explored in this context. This work focuses on the influence of alongshore topography
and hydrogeological factors on geotechnical impacts near the shoreline owing to ocean surges driven by coastal storms, which
are projected to intensify and become more frequent in the future (Chini & Stansby, 2012; Tebaldi et al., 2012). In particular,
the three-dimensional dynamics of surge-induced inundation and the resulting shore-parallel distribution of pore-pressure
gradients in sandy beach areas are not well understood. Specific questions addressed in this work are: (1) Can surge-induced
pore pressure changes promote sediment liquefaction of the uppermost sediment layers (<5 m), and which areas across the
beach are the most vulnerable? (2) What is the relationship between beach morphology and the spatio-temporal evolution of
pore pressure gradients? (3) How do the hydrogeological properties (hydraulic conductivity, groundwater recharge) of the
coastal system affect the potential for failure? Field evidence is presented for the effect of storm surges on coastal groundwater
heads (Section 2), a criterion is derived (Section 3) for momentary soil liquefaction for beach slopes with groundwater
discharge based on existing solutions (Briaud, 2013), and a model framework is described (Section 4) and used to simulate
surges in theoretical beach settings and to examine their effect on sediment stability (Section 5).

## 2 Field evidence for hydraulic head changes during storm surges

Groundwater observations collected every 10 min from October 2014 to November 2017 in 8 wells deployed across a 500-m
wide barrier island on the Outer Banks of NC, near the town of Duck (Figure 1a) indicate that coastal storm waves and surge
significantly affect the freshwater equivalent heads from the beach to more than 310 m inland of the beach (Housego et al.,
2018). The study period included 27 storm events (including 4 hurricanes) in which wave heights measured in 26-m water
depth (NDBC Station 44100) often exceeded 3.5 m, surge (NOAA tide gauge 8651371) was between 0.5 and 1.0 m, and 36-
hr-averaged (to remove fluctuations owing to tides and wind wave motions) shoreline water levels increased from about 0.6
to 2.4 m owing to surge and wave-driven setup (included in the simulated surge height). In response to the increased ocean
water levels, the groundwater level under the ocean dunes rose 0.5 to 2.0 m. For example, following the passage of Hurricane
Joaquin in 2015, which caused offshore wave heights of 4.7 m ( and <1 cm of rainfall), head levels under the ocean dunes and
25, 90, 160, and 310 m farther inland increased 1.6, 1.4, 1.2, 0.9, and 0.5 m above pre-storm levels, respectively (Figure 1b).
These and other storm-driven increases in head levels changed the direction of the hydraulic gradient from toward the bay
(inland) during calm conditions to toward the ocean during storms (compare black and red points in Figure 1b under calm
conditions with those during the storm). After the shoreline water level returns to pre-storm conditions, the water table behind
the dune remains elevated and groundwater discharges back out through the beach as the water table recovers. During the





storm, the horizontal location of the shoreline remained more than 10 m seaward of the dunes, and thus there was no inundation
from overtopping, which could increase groundwater levels even farther inland. Changes in hydraulic gradients, including the
effects of inundation, are investigated in Section 4 with a numerical model that does not mimic the conditions in this field site,
but is a generalized representation of coastal hydrogeological systems.

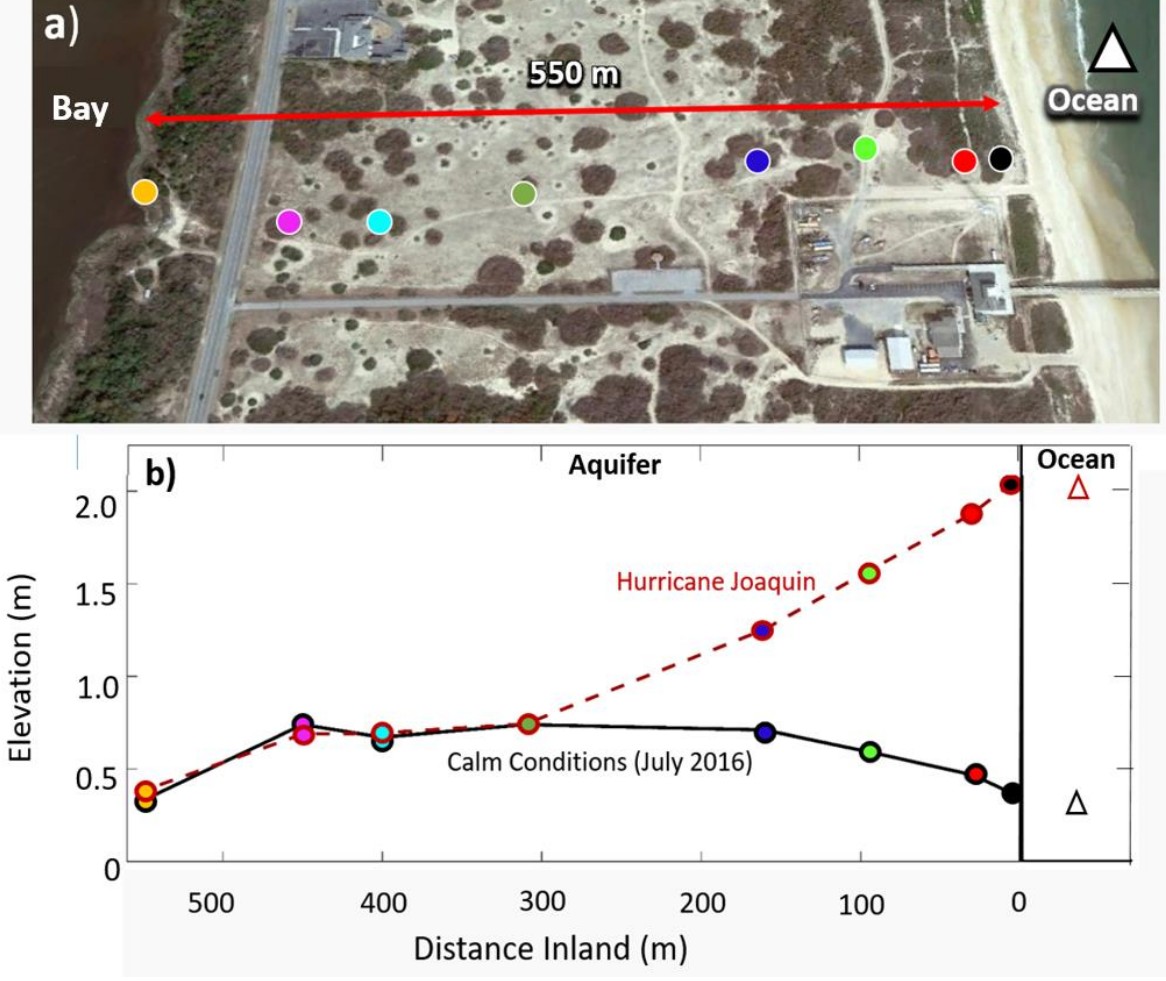


**Figure 1: a) © Google Earth image of the Outer Banks near Duck, NC, with the locations of groundwater wells (colored circles). (b) Elevation of the ocean level (triangles) and 36 hr-avg. freshwater equivalent groundwater heads (circles) vs. inland distance from the dune (x=0 m). Colors correspond to colors of symbols in (a)) for the average of the calm conditions in July 2016 (black triangle and circle outlines connected by black lines) and at the peak of Hurricane Joaquin (red triangle and circle outlines connected by red dashed lines).**

100



## 3 Conceptual model and governing equations

A conceptual model of a coastal system (Figure 2) includes infiltration of rain that recharges the aquifer with freshwater, resulting in fresh groundwater flow toward the ocean. In the nearshore area (typically within meters of the shoreline), an inclined freshwater-saltwater transition zone develops between the saline groundwater underlying the seafloor and the terrestrial fresh groundwater. The density gradient at the transition zone deflects the fresh groundwater flow upward, and produces focused groundwater discharge near the coastline that can be amplified by an order of magnitude or more relative to the average flow rate in the aquifer (Paldor et al., 2020). In phreatic aquifers, submarine groundwater discharge typically occurs within tens of meters of the coastline, depending on the recharge rates and aquifer properties (Bratton, 2010). In systems where the discharge is into a body of freshwater (e.g., a lake), the bottom of the lake is a constant head boundary, and thus the seepage is, by definition, perpendicular to the lakebed. This assumption is widely adopted in geotechnical calculations of groundwater discharge magnitudes. For example, in flow net solutions for classic dam and levee problems, the bottom of the river on both sides of the dam or levee is considered an equipotential line (Briaud, 2013). However, along the bottom of a saltwater body the freshwater-equivalent head is variable with bathymetry, and hence the seepage is not necessarily perpendicular to the seafloor and possibly represents a complex, three-dimensional problem with high spatiotemporal variability. To assess the risk of liquefaction in the context of the freshwater-saltwater transition zone and during coastal inundation events, the vertical component of the hydraulic gradient is computed to evaluate the potential for liquefaction (as will be derived in the following section) with the application of the variable-head boundary condition and the inclusion of variable-density flow solutions. It should be highlighted that in the current work, no effects of long-term loading and residual liquefaction were investigated. Hereinafter, the vertical hydraulic gradients will be discussed rather than the pore pressures or heads. In the next section the equations for soil failure potential in terms of the head gradients are derived based on previous derivations (Briaud, 2013). The magnitude of the hydraulic head gradient (Figure 2), which according to Darcy's law is the magnitude of the seepage vector divided by the hydraulic conductivity, is denoted i. Other variables used in the following calculations are shown in Figure 2 and summarized in Table 1.



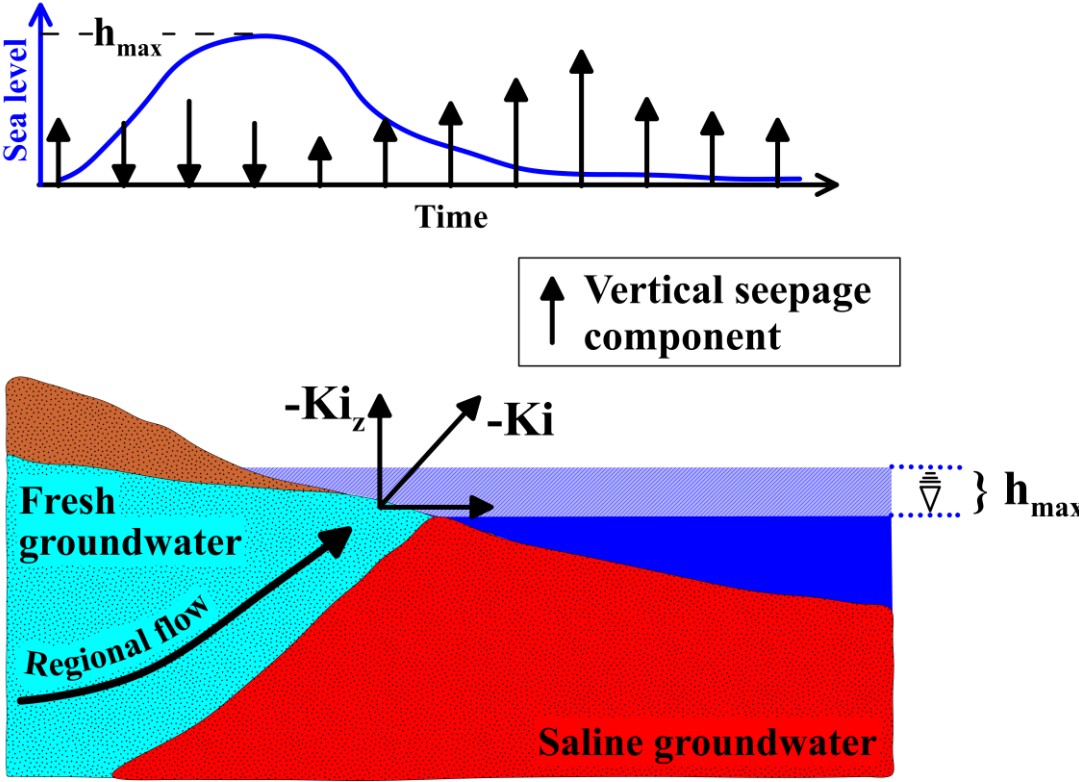

124

**Figure 2: A typical coastal hydrogeological system. Regional fresh (light blue) groundwater flows to the sea and upward due to variable-density flow along the freshwater-saltwater (red) interface. In the nearshore area, focused groundwater discharge occurs either into the sea (blue) or along a seepage face onshore. As shown in the top of the figure, when the surge begins, the direction of flow reverses (infiltration), and when the sea level reaches its maximal level (hmax) the surge retreats and the direction reverts back (exfiltration). The upward (positive vertical component) of flow reaches a maximum when the sea level is back to pre-surge level, before decaying to the steady-state magnitude.**















**Table 1: Variables used in the theoretical calculations and numerical simulations.**

| Parameter | Symbol | Value | Unit | Source |
|---|---|---|---|---|
| **Hydraulic conductivity** | K | 10-100 | m/d | Freeze & Cherry (1979) |
| **Anisotropy** | $K_x/K_z$ | 10 | | |
| **Seawater density** | $\rho_{sw}$ | 1025 | Kg/m$^3$ | |
| **Freshwater density** | $\rho_{fw}$ | 1000 | Kg/m$^3$ | |
| **Local water density** | $\rho_w$ | 1000-1025 | Kg/m$^3$ | |
| **Solid material density** | $\rho_s$ | 2650 | Kg/m$^3$ | |
| **Freshwater influx** | $q_0$ | 0.01-0.04 | m/d | |
| **Aquifer storativity** | $S_s$ | $10^{-4}$ | 1/m | Freeze & Cherry (1979) |
| **Porosity** | $n$ | 0.3 | | |
| **Longitudinal/Transverse Dispersivity** | $\alpha_L/\alpha_T$ | 1/0.1 | m | Gelhar et al. (1992) |
| **Maximum surge height** | $h_{0_{max}}$ | 3 | m | Chini & Stansby (2012) |


### 3.1 The criterion for liquefaction under groundwater seepage

Some publications distinguish between the terms "liquefaction" and "quick sand", with the former being used for earthquake-
induced fluidization of the soil, and the latter being related to failure due to upward flow (Briaud, 2013). However, the physical
meaning of the two is the same – geomaterial becoming weightless, which can result in erosion and sediment mobilization, or
loss of support of any infrastructure built into the soil. Here, the term liquefaction is used, although the analysis refers to surge-
induced changes in the subsurface flow rather than seismically induced flows. Following Briaud (2013), sand liquefaction
occurs when the pore pressure ($u_w$) at a certain depth ($z$) exceeds the total stress ($\sigma$), i.e. when the effective stress ($\sigma'$) goes to
zero:

$$\sigma' = \sigma - u_w \leq 0$$

(1)

Neglecting the possibility that gas is still trapped in the pores and assuming a submerged unit weight can be applied, the
criterion for localized, momentary liquefaction in inundated regions can be written in a gradient form (Goren et al., 2013), in
which the vertical pore pressure gradient (positive downward gradient generates upwards flow) exceeds the submerged unit
weight of the soil ($\gamma_{sub}$):



$$\gamma_{sub} + \frac{\partial u_w}{\partial z} \leq 0 \tag{2}$$

where

$$\gamma_{sub} = (1 - n) \cdot \left(\rho_s - \rho_{fw}\right) \cdot g \tag{3}$$


in which $\rho_s$ is the density of the beach material (sand), and $\rho_w$ is the density of the local water, which has a value between that
of seawater ($\rho_{sw} \approx 1025 \, kg/m^3$) and freshwater ($\rho_{fw} \approx 1000 \, kg/m^3$). This failure criterion is similar to Yeh and Mason
(2014), who studied liquefaction of a fully saturated sediment following a tsunami.
The constant value of porosity ($n$=0.3) is typical for sandy soils, but neglects localized variations in sand bulk density in the
simulated area. Furthermore, it is noted that the use of the submerged unit weight of soil is likely an underestimate of the actual
unit weight for soils under storm-surge conditions, since saturated conditions may prevail prior to inundation and the saturated
unit weight is higher than the submerged ($\gamma_{sub} = \gamma_{sat} - \gamma_{fw}$). However, this work aims to harness a hydrologic modeling
framework to assess the spatio-temporal distribution of surge-induced changes in hydraulic gradients. To that end, the
liquefaction assessment is limited to the effects of vertical pressure gradients, momentary liquefaction, and the application of
the submerged unit weight. It should be noted that studies have shown partially saturated sediments (e.g., in inundation areas)
are typically prone to momentary liquefaction (Mory et al., 2007; Yeh and Mason, 2014). Mory et al. (2007) showed that even
a 6% air content may alter the potential for momentary liquefaction. For the gradient-form criterion to hold, this condition
would need to be met continuously from the surface to the depth of the liquefied layer (Goren et al. 2013), as accounted for in
the analysis below.
Here, the momentary liquefaction criterion is related to vertical components of seepage vectors to compare the results of the
groundwater model with the failure criterion. The 3D model considered here (see below) could be used to examine the
horizontal components too, and to analyze the potential for shear failure, not only for momentary liquefaction (Zen et al.,
1998). However, for the sake of simplicity and in the interest of focusing on the questions addressed here, such an expansion
is not attempted in the current study. It would require further assumptions on the soil characteristics (internal friction, cohesion)
and a localized analysis of the local slopes for each point in the domain. According to Darcy's law the vertical flow velocities
($v_z$) are equal to the product of the (local) vertical head gradient and the vertical hydraulic conductivity $K_z$:

$$v_z = -K_z \left(\frac{1}{\rho_{fw}g} \frac{\partial u_w}{\partial z} + 1\right) \tag{4}$$


thus, the vertical pressure gradient becomes

$$\frac{\partial u_w}{\partial z} = -\rho_{fw}g \left(\frac{v_z}{K_z} + 1\right) \tag{5}$$




Substituting Equations 3 and 5 into Equation 2 yields:

$$(1-n)\cdot\left(\rho_s - \rho_{fw}\right)\cdot g - \rho_{fw}g\left(\frac{v_z}{K_z} + 1\right) \leq 0 \tag{6}$$


From Equation 6, the value of the critical vertical head gradient ($i_c$) is that above which the effective stress is zero or less:

$$\left(\frac{v_z}{K_z}\right)_c \equiv i_c = (1-n)\cdot\frac{\rho_s - \rho_{fw}}{\rho_{fw}} - 1 \tag{7}$$

This result is similar to that derived by Briaud (2013), but here it is derived for saturated groundwater flow, which is the
appropriate formulation for the scenario of surge-induced changes in the groundwater flow regime. Using Darcy's law in this
context assumes that during the surge the groundwater flow remains largely laminar, which is likely for storm-surge conditions
and is a common assumption in similar studies(Abdollahi & Mason, 2020; Guimond & Michael, 2021; Paldor & Michael,
2021; Yang et al., 2013; Yu et al., 2016). For convenience, the magnitude of negative (destabilizing) vertical head gradients
which initiate positive vertical velocities, is hereinafter denoted $i_z$ and presented in positive values. Using typical values for
porosity, solid particle density, and freshwater density for beach material ( $n = 0.3$ ; $\rho_s = 2650\ kg/m^3$ ; $\rho_{fw} =$
$1000\ kg/m^3$, respectively), Equation 7 suggests the critical value of vertical head gradient is about $i_c = 0.15$. The following
analyses use this value as a threshold for liquefaction, with simulated values of $i_z$ normalized by the critical value $i_c = 0.15$
as the seepage-liquefaction factor (SLF):

$$SLF = \frac{i_z}{i_c} \tag{8}$$

In Equation 8, $i_z$ is the actual simulated or observed vertical head gradient, defined as $i_z = -\frac{v_z}{K_z}$ (Eq. 4) and $i_c$ is the theoretical
liquefaction threshold (Eq. 7). Thus, any point in space and time in which simulated SLF is close to 1 is potentially nearing
liquefaction. A layer in which SLF approaches 1 continuously from the surface to a depth $Z_l$ is considered a "critical layer" of
thickness $Z_l$.
**4 Hydrologic model**
The effect of storm surges on groundwater flow is simulated using Hydrogeosphere (HGS) – a 3D numerical code that couples
surface and subsurface flow and solute transport (Therrien et al., 2010). For the surface flow, HGS solves the Saint-Venant
equations (also known as nonlinear shallow water equations), and for the variably saturated subsurface flow it solves the
Richards equation. The salt transport equation is solved in its advective-dispersive form, and the variable-density flow solution
is coupled to the transport solution through a linear equation of state. Hydrogeosphere has been successfully employed to
simulate storm surges in several recent studies (Guimond & Michael, 2020; Yang et al., 2013, 2018; Yu et al., 2016), and here
it is applied to assess the risk for sediment liquefaction and erosion from surge-induced pore water head gradients. This is a
novel interdisciplinary approach, applying a robust 3D hydrologic model in the context of coastal geomechanics.





The model domain (Figure 3) is 4000 m (cross-shore, X) by 2500 m (alongshore, Y), extending to a depth of 30 m below the
mean sea level (Z=0). The terrestrial extent of the domain is 3550 m (450<X≤4000), with the ocean spanning 0≤X≤450 (Figure
3). The elevation at the ocean side boundary is Z(X=0)=-1, so the seafloor slope is 1/450≈0.0022. This slope is representative
of U.S. Atlantic and Gulf coastal systems averaged over large cross-shore distances (e.g., from the beach to the mid continental
shelf). Although local slopes in the surf and beach often are much steeper than those used here, this study is focused on the
liquefaction in and near the inundated dune system. The average surface elevation inland (X=4000 m) is 5 m, so that the
average land surface slope is 5/3550≈0.0014. Thus, there is a change in average slope at the coastline, as the offshore portion
is steeper (~0.0022) than the onshore (0.0014), as in many coastal areas. A simulation with a -0.5 m sea level (i.e., still water
shoreline at X=225 m) indicates that critical vertical hydraulic gradients occur near this change in overall slope irrespective of
the shoreline location (Figure A1 in the Appendices). A simulation with a larger beach slope (Z(X=0)= -
6;slope=6/450=0.0130) resulted in similar vertical hydraulic gradients as the baseline slope (0.0022) (Figure A2 in the
Appendices), indicating that although the baseline slope is lower than typical, the analysis based on it is also valid for steeper
slopes. The domain of the finite difference model consists of 44,000 rectangular cells, where the cell sizes in the X and Y
direction are 25 and 50 m, respectively. The cell size in the Z direction varies from 8 m in the bottom of the domain to about
0.5 m in the top 2 m to balance between computation time and the resolution necessary to resolve the dynamics close to the
surface (Figure 3). The homogenous hydraulic conductivity Kx is 50 m/d for the baseline simulation and Kx varied between
10 and 100 m/d in sensitivity analyses. In all simulations, the anisotropy was 10 (i.e., the vertical hydraulic conductivity, Kz,
was 10 times lower than the horizontal hydraulic conductivity, Kx). This range of hydraulic conductivity with a porosity, n, of
0.3 is typical for sandy beach environments (Freeze and Cherry, 1979). Although a change in K could be associated with a
change in n for some sediments and mixtures, due to the potentially complex relationships between porosity and the sediment
textural properties, including grain size distributions, shapes, and K, the porosity was kept constant in the simulations presented
here.



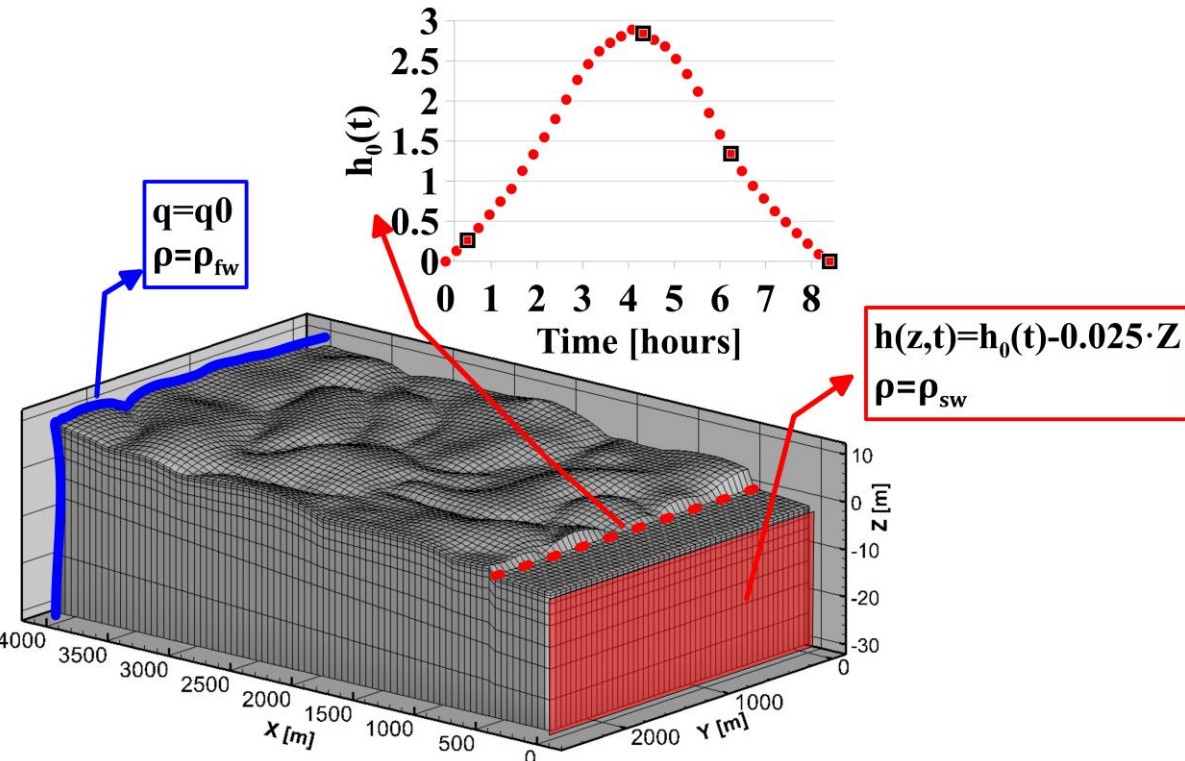

**Figure 3: Hydrogeosphere model domain as a function of the vertical Z, cross-shore X, and alongshore Y dimensions, boundary conditions (red and blue boxes), and the surge height evolution curve (inset). The blue curve is the terrestrial freshwater recharge boundary, the red rectangle is where a fixed seawater head and concentration are applied to the subsurface domain, and the red dashed line is where the sea level height boundary condition (h_0 (t)) is applied on the surface domain. For the steady-state simulations h_0 (t)=0, and for the transient simulations the curve in the inset is applied. The black squares in the inset mark the times plotted in Figure 5.**

The boundary conditions in the simulations were applied in two stages – a steady-state period and a transient surge period. For the steady-state simulations, terrestrial boundary conditions of constant freshwater specific recharge (q=q_0,ρ=ρ_fw) were applied on the vertical wall at the inland edge of the subsurface domain at X=4000 (blue curve in Figure 3) (Ataie-Ashtiani et al., 2013; Yang et al., 2018; Yu et al., 2016). The opposite edge of the domain at X=0 (red wall in Figure 3) was a typical sea boundary condition with depth-dependent head and saline ocean water (h=-0.025·Z; ρ=ρ_sw). On the surface domain the only boundary condition is applied on the coastline X=450 m, red dashed line in Figure 3) as a fixed, time-dependent head (h=h_0 (t)) and seawater density (ρ=ρ_sw). The applied head on the coastline was held at zero through the steady-state simulations. For the transient surge simulations, the coastline head was varied over 8.5 hours between zero and a 3 m maximum surge height (inset in Figure 3). A sea level of 3 m above the mean represents a combined high-tide and surge event with a projected





return period of 100 yr by the year 2050 in the East Coast of the United States (Tebaldi et al., 2012). The ocean surface was
assumed to be spatially constant at any time, and effects of wind waves were not simulated.
The sensitivity of the results to the topography and hydrogeologic parameters was tested, including freshwater influx (0.01<
$q\_0 < 0.04$ m/d, Figure 3 and Table 1) and hydraulic conductivity  (10 < Kx < 100 m/d, Table 1, typical values for sandy
beaches (Freeze & Cherry, 1979)). For the baseline hydraulic conductivity (Kx=50 m/d) the range of overall (land-to-sea)
hydraulic gradients, calculated as $q\_0/K\_x$ , was 0.0002 and 0.0008, on the lower side of typical coastal settings (roughly
around 0.0010), and so the calculated hydraulic gradients in the current analysis are considered a conservative estimate. Two
topographies (Figure 4) (Yu et al., 2016) were generated with ARCMAP 10.0 Geographic Information System (GIS) software
(ESRI, 2011), using multigaussian random fields that were transformed (Zinn & Harvey, 2003) to connect either topographic
highs or lows rather than the median topographic values as in the non-transformed multigaussian fields. The first topography,
named "River" (Figure 4a), is characterized by surface depressions that connect to the sea. The topographic lows are connected,
forming "river"-like patterns in the surface morphology), superimposed on the background slope of 0.0014. The second
topography, "Crater" (Figure 4b), features connected crests surrounding disconnected surface depressions, such that the highs
are connected, forming "crater" like shapes. The two topographies do not mirror each other (Figure 4), but represent reverse
alongshore trends near the shoreline (450<X<500 m) in which the area around 0<Y<300 m (2200<Y<2500 m) is the highest
(lowest) for the River topography and lowest (highest) for the Crater topography. Comparisons with real topographies of the
Delaware coastal plains (Yu et al. 2016) suggested that the River topography best represents real-world meso-topography.
However, the Crater topography provides important insights to how meso-topography controls the evolution of head gradients
during storm surges. In extreme flooding events (e.g., tsunami), large-scale changes in surface morphology (e.g., landslides)
may alter the pore-pressure distribution. These effects were excluded from the current work, as the simulated surface was
considered constant throughout the simulation. Additionally, soil deformation and the resultant stress re-distribution were not
considered in this model, as the hydrologic model (HGS) assumes constant porosity.

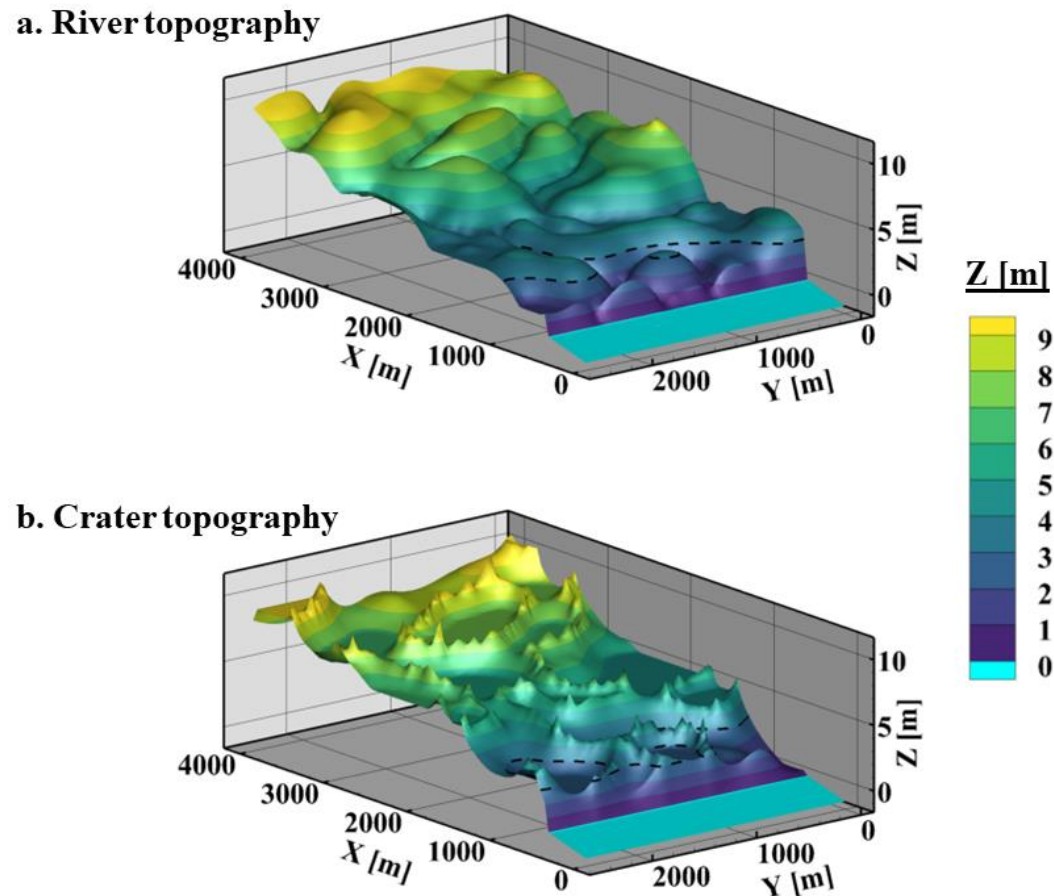


**Figure 4: (a) River and (b) Crater topographies as a function of the vertical Z, cross-shore X, and alongshore Y coordinates. Light**
**blue is the offshore bathymetry, and the coastline is at X=450 m. The overall slope accounting for macro-topography is the same for**
**both topographies, the average elevation at X=4000 m is ~5 m, making it a slope of 5/3550≈0.0014. The dashed black curve marks**
**the Z=3 m contour, which is equal to the maximum surge-induced sea level (hmax).**


For each simulation, the vertical hydraulic gradients ($i_z$ in Equation 8) are calculated over a vertical slice along the coastline,
i.e., the plane defined by X=450, and normalized by the threshold defined by Equation 7 ($i_c$) to calculate the SLF (Equation
8). As explained in Section 3 above, values of SLF that approach 1 are considered critical for liquefaction. When SLF≪1 the
simulated surface theoretically is stable. Only upward, destabilizing velocities (exfiltration) are considered, and so negative
velocities were assigned a value of $i_z=0$.



## 5 Results

The baseline case ('River' topography with $q\_0=0.02$ m$\lor$d ;$K\_z=5$ m$\lor$d) includes a 3 m surge and simulates the resultant changes in head gradients (Figure 5). During the inundation stage when sea level is increasing, the head gradients increase landward in front of the moving surge, and in the flooded zone there is infiltration (head decreases downward, $\nabla h>0$). After the peak of the inundation, when the high-water levels begin to recede, downward gradients (i.e., head increases downward, potentially destabilizing) develop underneath the still-water shoreline ($X=450$ m). These downward gradients increase in magnitude as the water level recedes, and the subsurface system relaxes back to background levels (not shown in Figure 5) within ~50 days for the high-K aquifers to ~500 days for the low-K aquifers, similar to prior simulations of storm impacts (Robinson et al. 2014). The peak alongshore variation of the vertical hydraulic gradients occurs at the end of the inundation ($t=8.4$ hr, Figure 5d). The vertical hydraulic gradients onshore of the inundation front during run-up (Figure 5b) develop in subaerial areas, and therefore the calculated SLF for these zones is based on the saturated unit weight ($\gamma\_sat=\gamma\_sub+\gamma\_fw$) of sediments rather than the submerged unit weight ($\gamma\_sub$, Equation 3), and the model-predicted liquefaction may not occur in real systems because saturated soils are more stable than submerged ones (Briaud, 2013).





**Figure 5: Surface inundation and vertical hydraulic gradients at (a) 0.5, (b) 4.3, (c) 6.2, and (d) 8.4 hr after the simulated surge begins (for the surge height at these times refer to Figure 3). In each panel, the surface domain is shown on top, the subsurface 3D domain and vertical gradients are shown below, and two cross sections through the subsurface are shown: shore-parallel (left in each panel) and shore-perpendicular (right). The locations of the sections are shown on the 3D plot as red dashed lines (for shore perpendicular) and yellow dashed lines (for shore parallel). The upper two panels are during the run-up stage and the lower are during the retreat stage. Refer to Figure 3 for the surge height at each time shown here. Note that downward gradients (head increases downward) are plotted as positive and upward gradients (head increases upward) are plotted as zero.**

The head changes (Δh in Figure 6) between the steady state and the peak of the inundation inversely follow the topography (black contours in Figure 6a and b). For the highest topographic elements (Y=0 m for the "River" and Y=2500 m for the

...





"Crater"), which are not inundated, the simulated heads are approximately equal to the maximum ocean level at the dune crest
(X ~ 460 m), and decay inland over ~100 m, roughly consistent with field observations (Figure 1). The maximum head changes
(purple colors in Figure 6a) inland of the shoreline (X >475 m) at peak surge occur in the inundated topographic lows. Toward
the end of the simulated surge (t=7.2 hr, Figure 6b) the surge-induced overpressures are released in the topographic lows (low
values of Δh in Figures 6b). The head differences also are low in the topographic highs because the heads there did not rise
significantly during inundation. In contrast, the intermediate topographic features show high head differences (dark purple in
Figure 6b). The lowest near-shore (450≤X≤500 m,900≤Y≤1200 m) topography undergoes similar head changes during the
peak surge for high and low K (compare Figure 6a1 with 6a3). However, in the low K case (Figure 6a3, 6b3), the heads are
not released effectively as the surge recedes, and significant excess heads of ~1 m difference remain near the end of the surge
(compare Figure 6b3 with 6b1 for X ~ 450 m).
When the surge has retreated (t=8.4 hr), the head gradients at the dune toe (initial shoreline) (X = 450 m) reach their maximum
(Figure 6c1-c3). In all simulations critical gradients (SLF→1, red zones in Figure 6 c1-c3) are simulated at some locations
below the shoreline, supporting the findings of several recent field studies in which momentary liquefaction was observed in
response to inundation events (Sous et al., 2016; Yeh & Mason, 2014). The alongshore distribution of the surge-induced
gradients is insensitive to the freshwater influx ($q_0$), even though the antecedent local hydraulic gradients differed by up to a
factor of 4 between simulations (Figure A3 in the Appendices, note that the values of the antecedent local gradients are about
an order of magnitude lower than the peak gradients). The depth and alongshore locations of the areas prone to liquefaction
(i.e., SLF ~ 1) are sensitive to the topography (compare Figures 6 a1,b1,c1 with a2, b2, and c2) and the hydraulic conductivity
(compare Figures 6 a1,b1,c1 with a3, b3, and c3). The two topographies exhibit a similar spatial pattern of SLF (Figure 6c1
and c2) even though the differences in topography (Figure 4) cause significant differences in the surge-induced head changes
(Figure 6 a1 and a2). For example, the area to the left of the domain (Y≤~300 m) is a topographic low in the Crater topography
and undergoes significant head changes at the peak of the inundation (Figure 6a2), whereas for the River topography there is
a topographic high for Y≤~300 m, which is not as strongly affected by the surge (Figure 6a1). However, in both cases this area
is where the least significant vertical head gradients develop (Figure 6c1 and c2).
The hydraulic conductivity has a significant effect on the simulated surge-induced gradients (Figure A4 in the Appendices).
Decreased hydraulic conductivity causes higher peak vertical gradients and changes the spatial (shore-parallel) distribution of
the gradients (compare Figure 6c3 with 6c1, especially near Y = 1000 m, and also see Figure A4). Furthermore, decreasing
hydraulic conductivity alters the depth $Z_l$ of "critical layers" with SLF = 1 (Equation 8) (compare Figure 6c3 with 6c1). In
the high-K simulations (Figure 6c1 and c2), the depth $Z_l$ of these "critical layers" with SLF ~ 1 ranges between 0 and 2.5 m,
and in the low-K simulation (Figure 6c3) $Z_l$ is up to ~5 m.



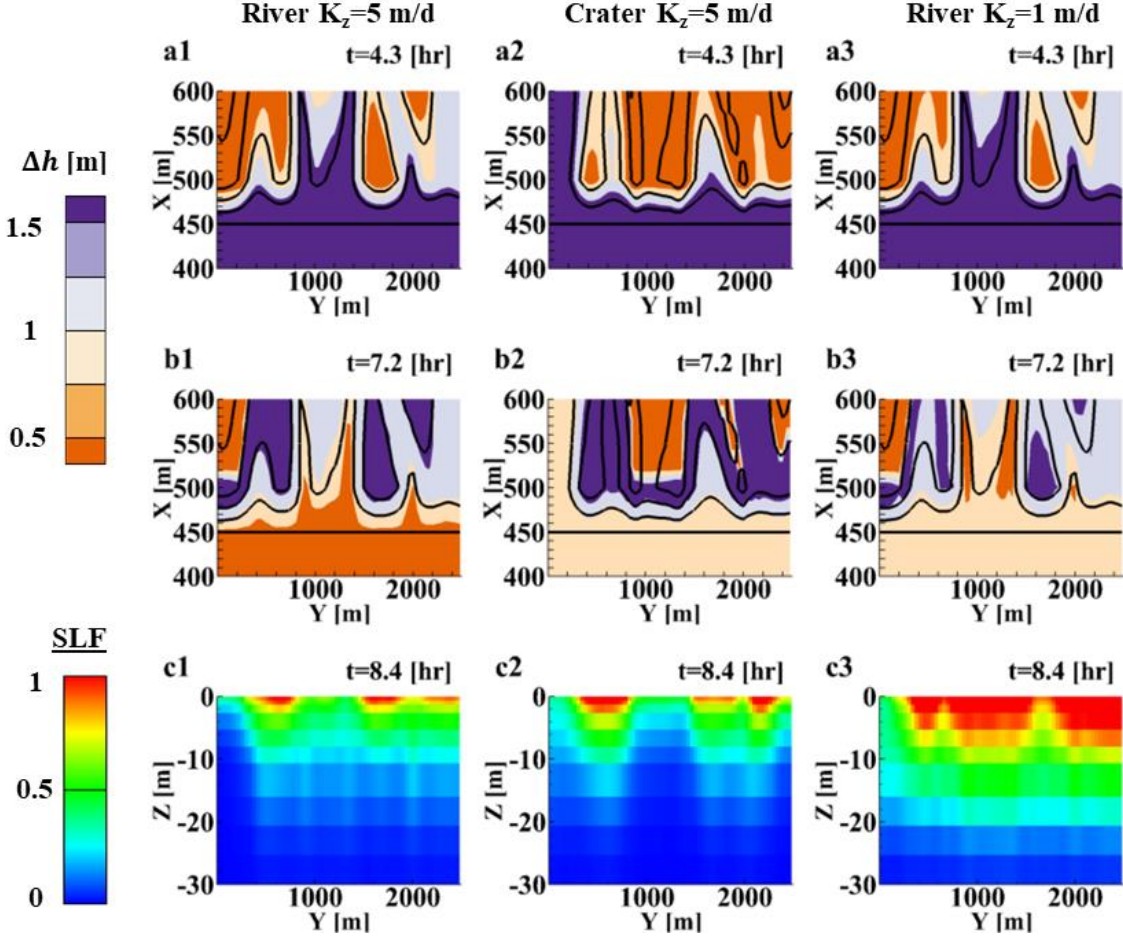

**Figure 6: Top row (a1-a3): maps of the maximum near-surface head differences between those at the peak of the inundation and the initial, pre-surge values (denoted Δh_1) as a function of the cross-shore X and alongshore Y coordinate. Middle (b1-b3): maps of the maximum subsurface head differences between those near the end of the surge (t = 7.2 hr, Figure 3) and the initial, pre-surge heads (denoted Δh_2) as a function of X and Y. Bottom (c1-c3): Liquefaction potential SLF at the shoreline, X = 450 m, as a function of the vertical Z and alongshore Y coordinate. These 3 metrics are plotted for River topography with Kz=5 m/d (left, a1-a3), Crater topography with Kz=5 m/d (center, a2-c2) and River topography with Kz=1 m/d (right, a3-c3). In the upper and middle panels (map views a1-a3 and b1-b3) the black contours are surface elevation with 1 m intervals. The horizontal line at X=450 is the coastline (Z=0). The lower panels are plotted for t=8.4 hr, the time at which the vertical gradients peaked in all simulations all along the coastline.**

The relationship between coastal topography and the surge-induced liquefaction potential is evident when comparing the surface elevations 50 m landward of the coastline (X=500 m) and the peak vertical gradients below the coastline for different topographies and K's (Figure 7). Here, the SLF=0.7 contour is used for statistical stability (there are more locations with SLF≥0.7 than with SLF=1). For both topographies, when K is high, SLF typically remains less than 0.7 (in Figure 7 where the

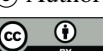



blue diamonds = 0) at the shoreline adjacent to the highest (Z > 3m) and lowest (Z < 1 m) topographic elements (marked by gray rectangles in Figures 7a and b), suggesting the intermediate topographic features may lead to the strongest vertical hydraulic gradients and liquefaction potential. However, the height of intermediate features that produce high gradients may be dependent on the site and hydrogeological parameters. For example, in the two simulations with higher Kz, 1-3 m topographic features are associated with most of the significant surge-induced gradients (Figure 7a and b). For the lower Kz case, significant gradients occur also below the lowest area (Figure 7c), and only the highest area that is not inundated does not develop significant gradients (gray rectangle in Figure 7c).

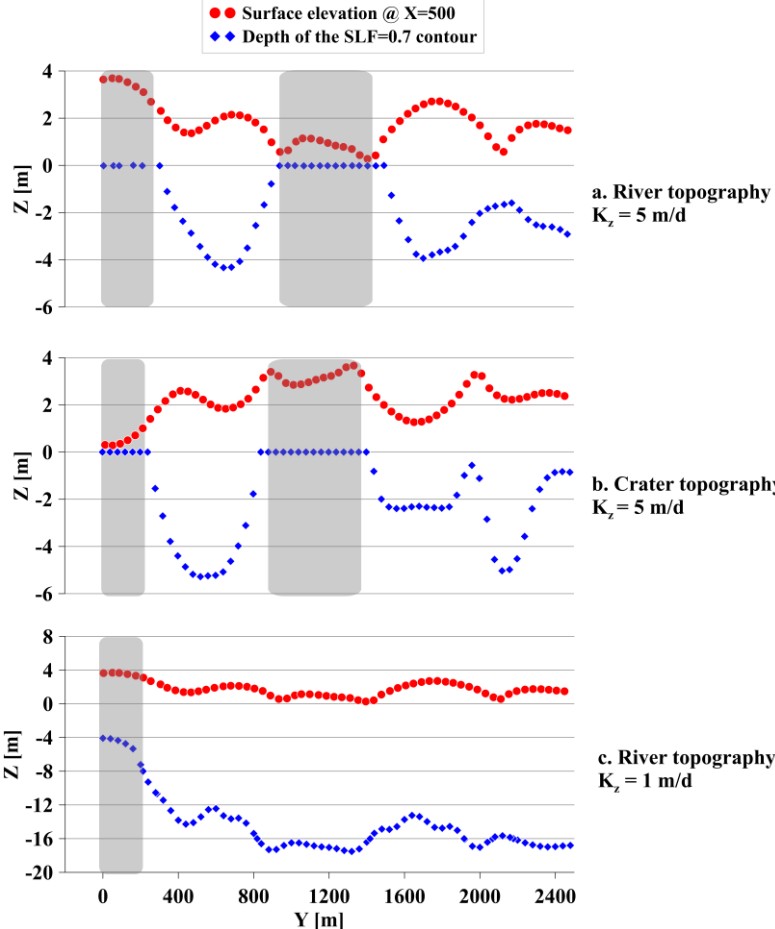

**Figure 7: Topographic elevation at X=500 m (50 m onshore of the shoreline, red circles) and depth of the SLF=0.7 contour below the shoreline (blue diamonds) versus alongshore coordinate Y for (a) the River topography with Kz=5 m/d, (b) Crater topography with Kz=5 m/d, and (c) River topography with Kz=1 m/d. Deeper locations of the SLF=0.7 contour (blue diamonds) mean thicker "critical layers". The places where no significant critical layer develops (i.e., the elevation of the SLF=0.7 contour is Z=0) are marked by gray rectangles.**





**6 Discussion**
**6.1 Alongshore variability**
The simulations suggest that alongshore variability of the magnitudes of the vertical gradients is strongly associated with the
coastal topography (Figures 5-7). To induce high gradients and deep critical layers when pressures are released, it is necessary
to have inundation resulting in high infiltration and increased heads. Thus, topographic highs that are not inundated cannot
develop high gradients (Figures 6 and 7). Meanwhile, overpressures often are released efficiently from inundated areas as the
surge recedes. Topographic elements that are low enough to be inundated, but are also high enough to limit the post-surge
exfiltration may prevent release of pressures, possibly explaining the correlation of liquefaction potential with intermediate
topographic features (1-3 m high for a 3 m surge). This explanation would suggest that the characteristic elevation of
"intermediate features" would scale with the surge magnitude. Pressure releases also can be limited by low hydraulic
conductivity. Thus, the simulations suggest the areas most susceptible to destabilization (i.e., deep critical layers) are those
where topography is low enough to be inundated widely, and high enough that the pressure release is limited. The range of
susceptible topographic elements depends on hydraulic conductivity, which also has a sweet spot of vulnerability: A simulation
with even lower hydraulic conductivity ($Kz=0.05$) showed that very low values of K limit the surge-induced infiltration and
thus critical gradients develop only to a limited vertical extent and the alongshore variability (i.e., the dependency on onshore
topography) diminishes (Figure A5 in the Appendices).
**6.2 Cross-shore spatiotemporal variability**
During the flooding stage, negative vertical gradients (infiltration) that do not promote sediment instability occur at and
seaward of the moving inundation front. Positive vertical gradients occur landward of the front (top right panel in Figure 5)
owing to alteration of the pre-existing steady-state flow field (Figure 2) by the advancing overpressures from the surge.
However, the simulated values of SLF=1 inland of the inundation front are not necessarily sufficient to liquefy the surface,
because the actual weight of the unsubmerged soil is greater than $\gamma\_sub$ (Equation 2). Nevertheless, the liquefaction potential
calculated here may still represent an underestimate, as Mory et al. (2007) showed that as little as 6% air content in the pores
may reduce the required pressure difference to liquefy the sediment by 0.01 m. This highlights the need to consider air contents
in future studies. Furthermore, these inland processes, and the potential for liquefaction in these areas, may be affected by
vegetation, trapping of gases, hysteresis of wetting and drying, and other processes that have not been considered here.
Nevertheless, the presented approach demonstrates the feasibility and a pathway to implement the concept of surge-induced
momentary liquefaction in a hydrological model that can predict variable-density groundwater flow in coastal and estuarine
environments.
The receding water levels after the peak of the surge allow fast release of the elevated heads that developed in the inundated
area, because the overlying burden of surge waters is removed abruptly. For all simulations at all alongshore locations, the
positive head gradients simultaneously reached a maximum when the water had receded completely (t=8.4 hr) and all the





inundation water overburden was released. The rate of head release determines the hydraulic gradients that occur in the soil
material, so that faster release of the overpressures produces lower positive head gradients. As the water recedes, the highest
release rates, and thus overpressures, develop under the beach area, where the slope changes from a terrestrial average slope
of 0.0014 to the seafloor slope of ~0.0022 (Figure 3). Thus, the simulations suggest the highest surge-induced gradients might
be expected under convex topography, for example near the berm or near a scarp in the beach face.
**6.3 Implications for coastal engineering**
Most previous studies of extreme wave-induced pressurization in coastal environments focus on cross-shore variability (Sous
et al., 2013, 2016; Turner et al., 2016; Yeh & Mason, 2014). Here, it is shown that under realistic hydrogeological conditions
(surge height, topography, groundwater flow regime – all based on values that are commonly observed in natural systems)
with alongshore varying topography there can be significant differences in storm-induced maximum hydraulic gradients and
in the depths of corresponding critical layers over small distances along the coastline (<500 m) (Figure 6). The simulations
suggest that beach and dune morphology are important factors determining the spatial variability of high gradients. Although
low-lying coastal areas may endure the greatest flooding, the largest hydraulic gradients and the deepest liquefaction layers
may occur at the toes of the intermediate-scale (1-3 m high for a 3 m surge) topographic features. While discussing practical
implications of the present analysis, it is important to remember that, as noted above, the model adopted here is a hydrological
model that does not explicitly simulate the soil dynamics and the surface and subsurface domains were assumed constant with
time through the simulations. This assumption overlooks other dynamic controls on the development of stresses, such as soil
deformation and surface erosion. Moreover, the analysis presented here isolates the vertical seepage component to calculate
the potential for momentary soil liquefaction. In a 3D framework, horizontal seepage components likely come into play and
other failure mechanisms, such as shear failure, are likely too (Zen et al., 1998). However, for the conclusions drawn here
regarding the spatio-temporal distributions of surge-induced gradients, the hydrologic modeling provides an important tool to
study the hydrogeological aspect of the problem. The model could be further expanded to include other components in future
work.
**7. Conclusions**
Field measurements from Duck, North Carolina, show that during Hurricane Joaquin the groundwater flow regime at the ocean
side was impacted substantially, and the hydraulic head gradient reversed its direction, followed by a period of recovery during
which downward gradients (upward fluxes) were regenerated. This suggests that hydraulic gradients generated by storm surges
may substantially affect the stability of beach surfaces. We explored this idea and its generality by harnessing a robust
hydrological model to simulate a generalized coastal system and found that in the nearshore area, surge-induced hydraulic
gradients may peak to critical levels that could potentially induce sand liquefaction. The locations where these critical gradients
occur are transient and depend on the beach morphology and hydraulic conductivity. Both the elevation of topographic features



and their permeability are important factors in promoting liquefaction. Elevations must be low enough to become inundated,
and high enough to retain elevated heads needed to build critical gradients. Similarly, hydraulic conductivity must be high
enough to allow floodwater to infiltrate, but low enough that water is not drained immediately such that critical gradients can
persist. This alongshore variability has not been observed in field measurements because the common approach in field studies
is to measure the cross-shore variability of hydraulic heads during storms. Importantly, this work presents a novel approach to
bridge the gap between coastal hydrology and coastal engineering, incorporating robust hydrogeological modeling in a
geotechnical framework.



**8. Appendices**

**Figure A1: Contours (color scale on the right) of peak SLF (t=8.4 hr) as a function of the vertical Z, cross-shore X, and alongshore Y coordinate for (a) a simulation with the coastline at -0.5 m ($X = 225\ m$) and (b) a simulation with the coastline at 0 m ($X = 450\ m$). The dashed black lines mark the coastline in each respective simulation. The slice with high SLF values in (a) is not underneath the simulated coastline.**

439



**Figure A2: Contours (color scale on the right) of peak SLF ($t = 8.4\ hr$) for a simulation with (a) bathymetric slope of $\frac{1}{450} \approx 0.002$ and (b) a simulation with a higher bathymetric slope ($\frac{6}{450} \approx 0.013$). The upper part of each panel shows the surface with the inundation water and the lower part is the vertical slice with the SLF values below the coastline (X=450 m).**



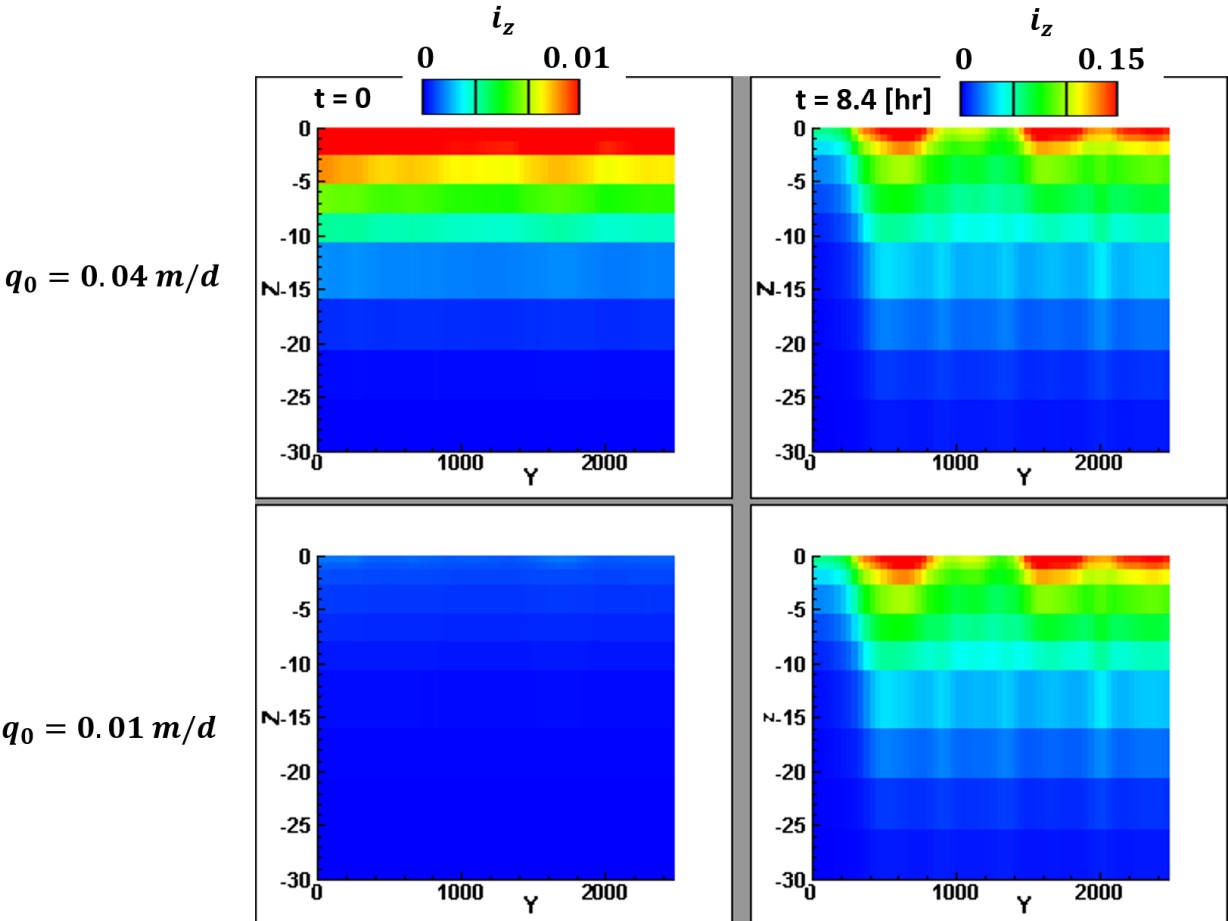

446

**Figure A3: Contours (color scales on the top) of vertical hydraulic gradients ($i_z$) at X = 450 m (shoreline location) for**

**the pre-surge conditions (left) and the end of the surge when gradients are maximum (right) as a function of vertical Z**

**and alongshore Y coordinates. Note the different color scales between the pre-surge (left) and the peak (right) plots.**

450







**Figure A4: Contours (color scale on the left) of peak SLF (t=8.4 hr) vertical slices at the shoreline (X = 450 m) for Kx and Kz of (a) 100 and 10, (b) 50 and 5, (c) 25 and 2.5, and (d) 10 and 1 m/d.**

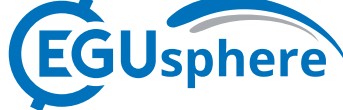



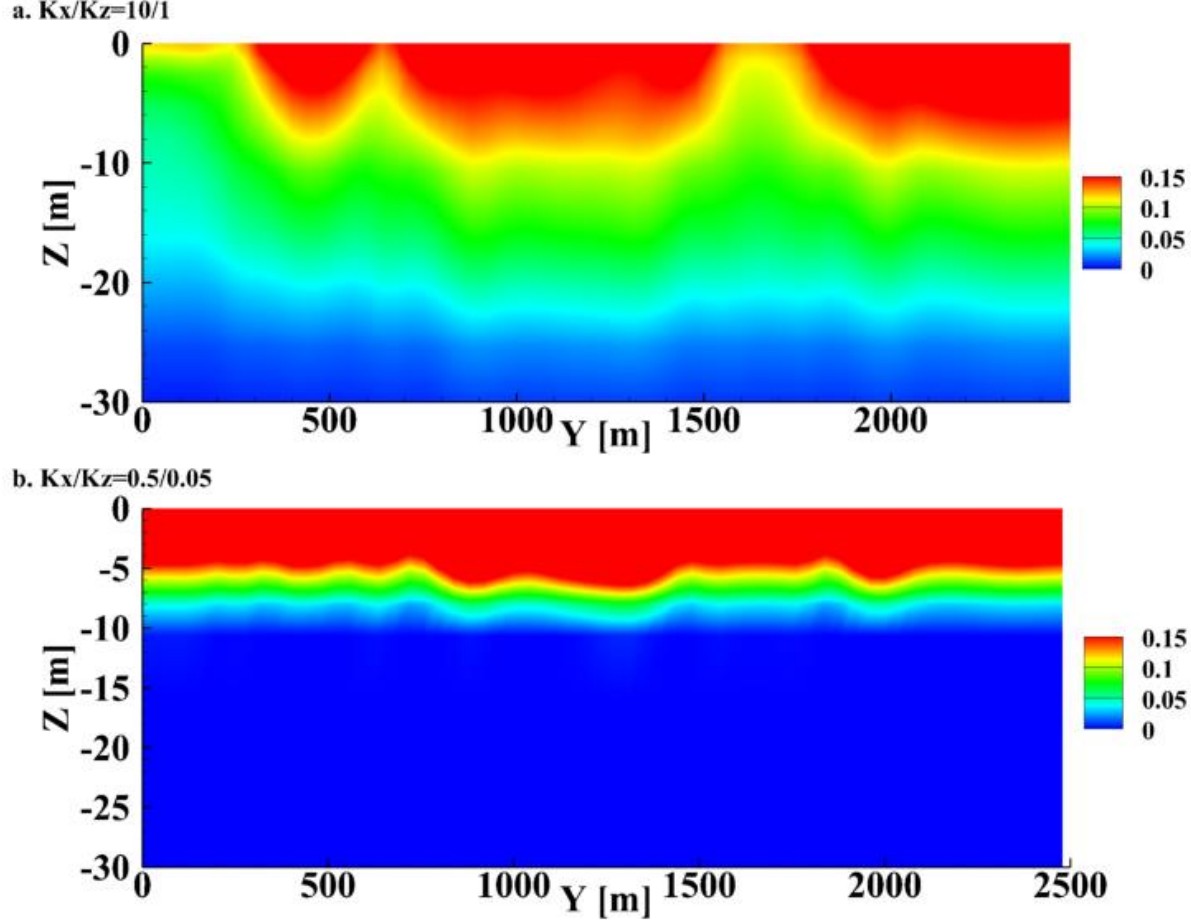

**Figure A5: Contours (color scales on the right) of the maximum vertical hydraulic gradients ($i_z$) at X = 450 m (shoreline location) for (a) $K_z = 1$ and (b) $K_z = 0.05$ ) as a function of vertical Z and alongshore Y coordinates.**

**Author contribution**

AP: conceptualization, investigation, visualization, formal analysis, writing (original draft); NS: conceptualization, formal analysis, writing (review and editing), funding acquisition; MF: formal analysis, writing (review and editing); BR; conceptualization, formal analysis, writing (review and editing), funding acquisition; SE: conceptualization, formal analysis, writing (review and editing), funding acquisition; RH; Data curation, visualization, writing (review and editing); RF formal analysis, methodology; HM: conceptualization, formal analysis, writing (review and editing), supervision, funding acquisition, resources.



**Acknowledgments**
We thank the staff at the USACE Field Research Facility and the PVLAB field team for helping to deploy and maintain
groundwater wells. Funding was provided by The National Science Foundation (OCE1848650, OIA1757353, OCE1829136,
EAR1933010, CMMI-1751463, and a Graduate Research Fellowship), US Geological Survey (NIWR 2018DE01G), a
Vannevar Bush Faculty Fellowship, the US Coastal Research Program, and the Woods Hole Oceanographic Institution
Investment in Science Program.

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
