# Peer review of "Coastal topography and hydrogeology control critical groundwater gradients and potential beach surface instability during storm surges"

_EGUsphere, 2022_

## Referee Comment (RC1)

Review of "Hydrogeological controls on the spatio-temporal variability of surge induced hydraulic gradients along coastlines: implications for beach surface stability"

**Summary**

A numerical groundwater flow model is used to quantify the vertical hydraulic gradients within a coastal unconfined aquifer experiencing a storm surge event. A "liquefaction" potential index is used to characterize where and when upward groundwater fluxes could lead to moments of minimal effective stress on sediment grains. Critical values of this index are shown to exist near the intersection of flood waters and topography, and flooded areas with longer flooded conditions resulted in more extended critical conditions.

**General comments**

- 1. Quick sand is not the same thing as liquefaction. Quick sand describes a state of the material when it has no effective stress acting upon it. Liquefaction is the process of liquefying such a material with an outside force acting on it. Static liquefaction is possible (Sadrekarimi, 2014), but it also requires an outside force and describes a process. I do not believe that the term liquefaction is necessary for this study to be useful. Using incorrect and inaccurate terminology as the foundation is not useful. Quick sand more accurately describes the state being modeled, but perhaps that label is unwanted. No label is needed once the state is described. It may be most accurately termed a "colloidal solution". The state could be connected to the many potential hazards it can respond to/cause with external factors considered, including liquefaction, piping, and uplift. Groundwater modeling has been performed to understand these effects using the "Factor of Safety" ratio, which is 1/SLF as defined in this study for coastal settings (e.g., Yang & Tsai, 2020) as well as numerous applications to levees (as already referenced in the manuscript). Liquefaction potential related to sea-level rise causing higher water tables (i.e., changes in saturation) require specific earthquake events (e.g., Grant et al., 2021 and references therein). Including vertical head gradients could provide additional information to these analyses, but none of this was developed in the current manuscript. (Aside: A commentary on how sea-level rise could lead to more areas at risk of high head gradients with surges could be added to the implications discussion section).
- 2. Overpressures and excess pressures are not used in a context that I am familiar. This generally implies greater than hydrostatic pressure (i.e., a confined or artesian pressure). I do not believe this is what is intended. Rather, I interpret these terms to mean raised water levels from the surge event that then need to drain away. The text does not make this clear, but it affects one of the main conclusions on the "intermediate" elevations playing an important role in setting low effective stress conditions.
- 3. I do not find the "Field Evidence" portion of the manuscript to contribute to the overall purpose of the study. It appears to be based on previously published results (Housego et al., 2018). Referencing this study and moving on within the introduction to set the hydrogeologic context seems sufficient. If the model were more directly connected to the field site, then retaining such information could be useful. The conclusion that storm surge head gradients at the field site "substantially affect the stability of beach surfaces"

is conjecture and no data are presented on beach stability or vertical hydraulic gradients (at least based on the information provided in Figure 1b). The response of the water table to a storm surge does not provide information on vertical gradients.

- 4. The difference between the submerged and saturated unit weights are not sufficiently clear to make them distinguishable other than that they are somehow different. Part of the problem is that the  $\gamma$ sat and  $\gamma$ fw are never defined. It is unclear how  $\gamma$ sub would be different from  $\gamma$ sat the water level or pressure is not part of the equation, so it should not matter if the land surface is submerged or saturated. What is different about these two terms other than something to do with freshwater? Is  $\gamma$ fw = 1? All of these unit weights need to be defined more clearly. I assume that these values depend on the modeled porewater salinity?
- 5. Varying the hydraulic conductivity a few times and considering two synthetic topographies represents a meager exploration of the parameter space. Interestingly, the parameters listed in Table 1 do not match the values used in the results figures and no justification for the ranges are provided (e.g., K in the 10-100 m/day range from Freeze and Cherry (1979) with no indication of sediment type and then a few values of Kz are reported in the later figures). How many models were performed? If only a few for each, then listing the exact values of the parameters seems feasible and useful. Using only one value of porosity for the critical index value seems overly simple. At a minimum, it could be useful to present what the range of the SLF index is for realistic coastal geomaterials.
- 6. Conceptually, I feel like the main conclusion that "topography" is important for setting the colloidal solution development is not fully explored. No alternative hypotheses or more nuanced hydrologic explanations are tested, even though the modeling produces the outputs to test them. The elevation of the land surface and ability to drain to the coast are invoked as the primary driver of the "intermediate" elevation cases where the lingering higher pressures occur. This is reasonable to me, but I believe the mechanism is more related to two controls 1) the initial water table depth setting the infiltration capacity and rate(s) as seen with tsunamis and storm surges (Cardenas et al., 2015), and 2) the horizontal vs vertical gradients allowing the pressure to dissipate in those areas. Despite the relative complexity of the synthetic topographies, it should be possible to normalize model inputs and outputs and explore these controls more deeply.

| Lines | Comment                                                                                                                                                        |
|-------|----------------------------------------------------------------------------------------------------------------------------------------------------------------|
| Title | I do not find that the title explains the study well. "Topographic controls on surge-induced critical groundwater gradients and potential surface instability" |
| 24    | Revised to explain what the revision should be "to include steeper portions of"                                                                                |
| 26    | Туро                                                                                                                                                           |
| 37    | Pressure distributions do not induce surface failure. An outside force/stress is required to induce this.                                                      |

**Specific comments**

| 66     | Flooding is generally a temporary event while inundation implies a longer-term      |
|--------|-------------------------------------------------------------------------------------|
|        | water coverage. I believe all uses of "inundation" in this manuscript refer to      |
|        | flooding.                                                                           |
| 86     | Add arrows to Figure 1 (if keeping)                                                 |
| 89     | For how long? Sense of time in Fig 1 is mostly missing.                             |
| 90     | Dune location needed in Fig 1a                                                      |
| 92     | Then why is this field study here? Not even one example run could be done to        |
|        | connect these observations with the SLF results?                                    |
| 121    | Seepage vector needs to be defined and developed more clearly. Not the same         |
|        | terminology as used later in either the methods or results. Assuming seepage        |
|        | means "specific discharge vector"? Why the magnitude? Isn't direction               |
|        | important? Figure 2 could be made more consistent with the                          |
|        | methods/equations/variables. Where is this seepage vector being calculated          |
|        | (especially for Fig 2)? At the top of the free surface? At some depth? Figure 2     |
|        | also does not show the "magnitude of seepage component" but the actual              |
|        | seepage component with changing directions.                                         |
| 144    | Quick sand and liquefaction are related but exceptionally different. This is a      |
|        | weak justification. This analysis does not study liquefaction.                      |
| 146    | Sand is not weightless ever. It has mass. "Suspended" and liquid-like or            |
|        | colloidal-like, sure. Inaccurate terminology.                                       |
| 152    | Full saturated or flooded?                                                          |
| 161    | What simulated area? In the generic models. Suggests a field site.                  |
| 163    | The use of however here implies there is a justification preceding it, but there is |
|        | not. What is the "however" referring to? What is or is not being done in the        |
|        | study? A new paragraph may be helpful or a rearrangement of the two sentences       |
| 1.00   | that establish the differences.                                                     |
| 168    | Enhance/increase/etc (alter in which way?)                                          |
| 178    | Vz is the specific discharge according to this equation. Seepage velocity is easily |
|        | confused with average linear velocity, and I do not see porosity in the equation.   |
| 105    | Providing units could also be useful to distinguish with the volumetric form.       |
| 185    | And how are they different? A sentence is needed to explain the difference          |
| 100    | rather than the existence of a difference.                                          |
| 189    | Positive and negative are confusing and dependent on the reference frame. When      |
|        | possible, it would help the reader to use upward and downward to describe flow.     |
|        | It would also be useful to minimize the use of gradient directions and stick with   |
| 101    | one convention that is clearly explained.                                           |
| 191    | Would be simple to provide a range of values                                        |
| 206    | Please develop now this is a novel interdisciplinary approach within the context    |
|        | of the previous work (e.g., Yang and Isal, 2020) and levee-based studies. This is   |
|        | the analysis of the outputs of a well-used groundwater model tool within a          |
|        | statement is a gross and unjustified over call                                      |
| 214 10 | Statement is a gross and unjustified over-sell.                                     |
| 214-19 | with are these results sentences in the methods? Seem out of place.                 |
| 238    | Are crater topographies a standard coastal type? "Closed-dune" or similar would     |
|        | get the point across without invoking the extraterrestrial.                         |

| 275       | Doesn't Figure 5 show 3D volumes of the SLF? The vertical slices appear to be                                                                              |
|-----------|------------------------------------------------------------------------------------------------------------------------------------------------------------|
|           | a visualization technique rather than "the only place we have results".                                                                                    |
| 276       | Provide the value of the threshold in this sentence and remove the next.                                                                                   |
| 291       | Unit weights difference still uninterpretable                                                                                                              |
| 308       | Not overpressure.                                                                                                                                          |
| 309       | Which head differences? In space or time? From what to what?                                                                                               |
| 313       | Not excess head                                                                                                                                            |
| 328       | And why does this matter?                                                                                                                                  |
| 348       | Justification is "because there's more of them"? Really? The nice idea with the                                                                            |
|           | Factor of Safety (1/SLF) is that you don't want to be exactly at the critical value                                                                        |
|           | but need some buffer. Explaining this choice within a FoS framework would                                                                                  |
|           | provide stronger justification.                                                                                                                            |
| 370       | What prevents this release? Slow drainage? Long flow paths? Other?                                                                                         |
|           | Developing this would allow the next statement on something explaining the                                                                                 |
| 27(       | similarities (correlation? statistical danger word)                                                                                                        |
| 3/0       | Implications for real systems with more anisotropy and heterogeneity could be
developed here with reference support on the importance of such things in |
|           | coastal groundwater hydrodynamics                                                                                                                          |
| 383       | Arguably, none of these results suggest liquefaction, but it does seem important                                                                           |
| 565       | to develop this further to explain why intruding flood waters with the same                                                                                |
|           | pressure gradients lead to different results                                                                                                               |
| 386       | How does this pressure (really pressure head) difference relate to those modeled                                                                           |
| 200       | in this study? Would help provide context of 0.01 m to an actual range – does                                                                              |
|           | this matter or not?                                                                                                                                        |
| 394       | This result is not presented – the "simultaneously reached". Only snapshots of                                                                             |
|           | model results are show, and no time-dependent results are provided to support                                                                              |
|           | this portion of the discussion.                                                                                                                            |
| 396       | Indication that the flow rates related to infiltration rates are an important control                                                                      |
|           | and should be analyzed. This sentence also says that lowering pressure reduces                                                                             |
|           | the pressure gradient and is self-evident. Lowering a numerator does have that                                                                             |
|           | effect on a fraction.                                                                                                                                      |
| 404       | Maximum {vertical} hydraulic gradients. Directions needed.                                                                                                 |
| 421-2     | This was in no way a result of this study.                                                                                                                 |
| Figure 2  | A hypothetical system. Need consistency with methods. Missing subscript.                                                                                   |
| Figure 5  | Are the 3D color volumes the vertical hydraulic gradient or SLF? The last                                                                                  |
|           | sentence of the caption implies they are gradients, but there is no colorbar for                                                                           |
| D' | gradient.                                                                                                                                                  |
| Figure 7  | I suggest removing Figure / and incorporating these results into Figure 6, which                                                                           |
|           | would benefit from a contoured SLF value and topography included above the                                                                                 |
|           | aeptn silces in c).                                                                                                                                        |
|           |                                                                                                                                                            |

**References**

- Cardenas, M. B. B., Bennett, P. C., Zamora, P. B., Befus, K. M., Rodolfo, R. S., Cabria, H. B., & Lapus, M. R. (2015). Devastation of aquifers from tsunami-like storm surge by Supertyphoon Haiyan. *Geophysical Research Letters*, 42(8), 2844–2851. https://doi.org/10.1002/2015GL063418
- Grant, A. R. R., Wein, A. M., Befus, K. M., Hart, J. F., Frame, M. T., Volentine, R., et al. (2021). Changes in Liquefaction Severity in the San Francisco Bay Area with Sea-Level Rise. In *Geo-Extreme 2021* (Vol. 0, pp. 308–317). Reston, VA: American Society of Civil Engineers. https://doi.org/10.1061/9780784483695.030
- Sadrekarimi, A. (2014). Static liquefaction-triggering analysis considering soil dilatancy. *Soils and Foundations*, *54*(5), 955–966. https://doi.org/10.1016/j.sandf.2014.09.009
- Yang, S., & Tsai, F. T.-C. (2020). Understanding impacts of groundwater dynamics on flooding and levees in Greater New Orleans. *Journal of Hydrology: Regional Studies*, 32, 100740. https://doi.org/10.1016/j.ejrh.2020.100740

---

## Referee Comment (RC2)

Review of
**Hydrogeological controls on the spatio-temporal variability of surge-induced hydraulic gradients along coastlines: implications for beach surface stability**

by Paldor et al.

This paper reports on a series of numerical simulations to qunatify surge-induced vertical hydraulic gradients in coastal aquifers. The topic is of great interest, the paper is well written and organized. However, I have several concerns before I can recommend the paper for publication in HESS.

1 – l.32: I do not interpret the beach groundwater observations of Sous et al. 2016 as soil failure, please check.

2- l.34: I do not fully agree with the definition given for liquefaction. A zero-stress soil needs an external force to be liquified

3- l.42: depends

4- l. 53: Mory et al. 2007, I would emphasize here that liquefaction events were related to the presence of a rigid structure in the soil, and rather use Michallet et al. 2009 (JGR) for the same site but finer analysis.

5- l.121: What is meant by "seepage vector" ?

6- I do not see the input provided by Section 2. The data analysis has been already published, and the results presented here do not bring real insight (no vertical gradient, nothing new than much older works) and certainly do not show the statement in the Conclusions section ('may substantially affect…')

7- l.163: Please detail the definition of unit weights and more generally provide a unified and clear discussion about saturation vs submersion effets (e.g. l. 292).

8- Can you justify the anistropy in K ?

9- Can you describe in detail your sensitivity analysis (parameters and ranges) ?

10- The surge imposed here shows the same typical height and time scales than typical macrotidal areas. Does it mean that the potential "liquefaction" predicted here can be observed in any comparable macro-tidal coast ? Please comment.

11- l.308, 368 etc: What is meant by "overpressure" ?

12- The role played by horizontal gradients is not explored, and this may significantly affect the interpretation.

---

## Author Comment (AC1)

We have received the first referee's comments (RC1) on our paper, titled: **Hydrogeological controls on the spatio-temporal variability of surge-induced hydraulic gradients along coastlines: implications for beach surface stability** (A. Paldor et al.). We thank the referee for their valuable feedback, which we believe will improve the revised paper. Below is a list of the referee's comments (in red) and our respective replies (in black). Text cited from the manuscript is in black italicized, with suggested revisions highlighted in blue with tracked changes.

**General comments:**

1) Quick sand is not the same thing as liquefaction. Quick sand describes a state of the material when it has no effective stress acting upon it. Liquefaction is the process of liquefying such a material with an outside force acting on it. Static liquefaction is possible (Sadrekarimi, 2014), but it also requires an outside force and describes a process. I do not believe that the term liquefaction is necessary for this study to be useful. Using incorrect and inaccurate terminology as the foundation is not useful. Quick sand more accurately describes the state being modeled, but perhaps that label is unwanted. No label is needed once the state is described. It may be most accurately termed a "colloidal solution". The state could be connected to the many potential hazards it can respond to/cause with external factors considered, including liquefaction, piping, and uplift. Groundwater modeling has been performed to understand these effects using the "Factor of Safety" ratio, which is 1/SLF as defined in this study for coastal settings (e.g., Yang & Tsai, 2020) as well as numerous applications to levees (as already referenced in the manuscript). Liquefaction potential related to sea-level rise causing higher water tables (i.e., changes in saturation) require specific earthquake events (e.g., Grant et al., 2021 and references therein). Including vertical head gradients could provide additional information to these analyses, but none of this was developed in the current manuscript. (Aside: A commentary on how sea-level rise could lead to more areas at risk of high head gradients with surges could be added to the implications discussion section).

We agree with the reviewer that quicks and may be a more suitable term, and have mentioned this at the beginning of section 3.1. In the revision we will change all instances of *liquefaction* to *quicks and* as the reviewer suggests (in the responses below, suggested revisions will include these replacements). As for the reference to Yang and Tsai (2020), this is a very useful paper and we thank the reviewer for drawing our attention to it. We will include in the revision references to this study in the Introduction and in the Methods (when deriving the SLF, which as the reviewer correctly notices, is 1/FS as defined there). We will also discuss our modeling results in section 6.3 in comparison with the results Yang and Tsai (2020) obtained for the Greater New Orleans area.

2) Overpressures and excess pressures are not used in a context that I am familiar. This generally implies greater than hydrostatic pressure (i.e., a confined or artesian pressure). I do not believe this is what is intended. Rather, I interpret these terms to mean raised water levels from the surge event that then need to drain away. The text does not make this clear, but it affects one of the main conclusions on the "intermediate" elevations playing an important role in setting low effective stress conditions. We agree, using the term over/excess pressure may be confusing with artesian conditions which are not considered here. We indeed mean groundwater pressures that are enhanced compared to calm conditions (i.e., surge-induced higher pressures). In the revision we will change all over/excess pressures to increased pressures.

3) I do not find the "Field Evidence" portion of the manuscript to contribute to the overall purpose of the study. It appears to be based on previously published results (Housego et al., 2018). Referencing this study and moving on within the introduction to set the hydrogeologic context seems sufficient. If the model were more directly connected to the field site, then retaining such information could be useful. The conclusion that storm surge head gradients at the field site "substantially affect the stability of beach surfaces" is conjecture and no data are presented on beach stability or vertical hydraulic gradients (at least based on the information provided in Figure 1b). The response of the water table to a storm surge does not provide information on vertical gradients.

We agree. In the revision we intend to remove the field observations (Section 2 and Figure 1) from the manuscript, and add reference to the published data instead.

4) The difference between the submerged and saturated unit weights are not sufficiently clear to make them distinguishable other than that they are somehow different. Part of the problem is that the ysat and yfw are never defined. It is unclear how ysub would be different from ysat – the water level or pressure is not part of the equation, so it should not matter if the land surface is submerged or saturated. What is different about these two terms other than something to do with freshwater? Is  $\gamma fw = 1$ ? All of these unit weights need to be defined more clearly. I assume that these values depend on the modeled porewater salinity?

To address this comment, we will edit the explanation given for these two quantities (after equation 3): Furthermore, it is noted that the use of the submerged unit weight of soil is likely an underestimate of the actual unit weight for soils under storm surge conditions, since saturated conditions may prevail prior to inundation and the saturated unit weight is higher than the submerged ( $\gamma_{sub} = \gamma_{sat} - \gamma_{fw}$ ). The use of  $\gamma_{sub}$  as the representative unit weight of simulated soil is appropriate for soils that are fully submerged, as it accounts for the buoyancy effect, considering the unit weight of the overlying water column ( $\gamma_w$ ). However, for the parts of the model landward of the inundation line, the saturated unit weight may be more suitable. This means that adopting  $\gamma_{sub}$  uniformly may be an underestimate of the actual unit weight in real systems ( $\gamma_{sub} = \gamma_{sat} - \gamma_w$ )

**We will also add these to Table 1.**

5) Varying the hydraulic conductivity a few times and considering two synthetic topographies represents a meager exploration of the parameter space. Interestingly, the parameters listed in Table 1 do not match the values used in the results figures and no justification for the ranges are provided (e.g., K in the 10-100 m/day range from Freeze and Cherry (1979) with no indication of sediment type and then a few values of Kz are reported in the later figures). How many models were performed? If only a few for each, then listing the exact values of the parameters seems feasible and useful. Using only one value of porosity for the critical index value seems overly simple. At a minimum, it could be useful to present what the range of the SLF index is for realistic coastal geomaterials.

We agree that further exploration of the parameter space would be useful and interesting. However, this study is aimed to show the importance alongshore topographic variability and to propose an approach for better tying hydrogeologic and geomechanical modeling/measurements. To that end, a more detailed

exploration of the parameter space may be off point. In the revision, we will stress this valuable point at the end of section 4 with the following text:

It is noted that exploring 4 values of hydraulic conductivity and two types of synthetic topographies may be a limited representation of natural systems. For example, Xu et al. (2016) showed that topographic connectivity is a dominant factor in the vulnerability of coastal aquifers to storm surge salinization, and we consider here only two of the topographies simulated there. However, the tested topographies and conductivities in this work serve as a preliminary exploration of hypothetical conditions that are likely representative of many natural systems, but is certainly not inclusive.

As for the hydraulic conductivities, we state the values and their suitability for beach sediments in lines 222-228, and also the Kz values. Additionally, Figure A4 in the appendices details the values simulated. To address this comment and to better clarify, we will revise as follows:

The homogenous hydraulic conductivity Kx is 50 m/d for the baseline simulation, and values of Kx = 10, 25, 100 m/d were also simulated as part of a sensitivity analysis and Kx varied between 10 and 100 m/d in sensitivity analyses. In all simulations, the anisotropy was 10 (i.e., the vertical hydraulic conductivity, Kz, was 10 times lower than the horizontal hydraulic conductivity, Kx). This range of hydraulic conductivity with a porosity, n, of 0.3 is typical for sandy beach environments (Freeze and Cherry, 1979).

6) Conceptually, I feel like the main conclusion that "topography" is important for setting the colloidal solution development is not fully explored. No alternative hypotheses or more nuanced hydrologic explanations are tested, even though the modeling produces the outputs to test them. The elevation of the land surface and ability to drain to the coast are invoked as the primary driver of the "intermediate" elevation cases where the lingering higher pressures occur. This is reasonable to me, but I believe the mechanism is more related to two controls 1) the initial water table depth setting the infiltration capacity and rate(s) as seen with tsunamis and storm surges (Cardenas et al., 2015), and 2) the horizontal vs vertical gradients allowing the pressure to dissipate in those areas. Despite the relative complexity of the synthetic topographies, it should be possible to normalize model inputs and outputs and explore these controls more deeply.

We agree that it is important to acknowledge these potential controls and we will add the following text to section 6.1 (black italicized is from the original manuscript for reference, red italicized is the proposed addition):

Thus, the simulations suggest the areas most susceptible to destabilization (i.e., deep critical layers) are those where topography is low enough to be inundated widely, and high enough that the pressure release is limited. An important factor that likely plays a role in this relationship between intermediate topography and critical gradients is the horizontal gradient. In places where horizontal hydraulic gradients can develop, a more efficient dissipation of surge-induced pressures may be expected, and therefore critical gradients are less likely. This may explain the absence of critical hydraulic gradients are important also when considering other modes of surface instability, such as shear failure. To assess the potential for shear failure, a Coulomb criterion must be derived, which is beyond the scope of the current study. Another factor that is known to control the vulnerability to storm-induced instability is the antecedent groundwater level which controls the infiltration capacity of flood waters (Cardenas et al., 2015). This may explain the absence of critical hydraulic gradients. leaving an intermediate range of topographies that are susceptible to surge-induced critical gradients.

**Specific comments:**

7) I do not find that the title explains the study well. "Topographic controls on surge-induced critical groundwater gradients and potential surface instability"

We agree that the title should mention the topographic controls. We will revise to: *Coastal topography and hydrogeology control critical groundwater gradients and potential beach surface instability during storm surges*".

8) Line 24: Revised to ... - explain what the revision should be "to include steeper portions of ..." Agreed. We will change to: "...might need to be revised to include other topographic features."

**9) Line 26: Typo**

We will change especFially to especially.

**10) Line 37: Pressure distributions do not induce surface failure. An outside force/stress is required to induce this.**

We will revise that sentence to read:

At the coast, ocean (waves, surge, tides, inundation) and terrestrial (groundwater heads, precipitation, and overland flows) processes concurrently contribute to changing pore pressures in beach and nearshore sediments, and changes in pore pressure distributions and gradients could thus induce failure of the surface.

11) Line 66: Flooding is generally a temporary event while inundation implies a longer-term water coverage. I believe all uses of "inundation" in this manuscript refer to flooding.We will change all inundation to flooding, though the terms are often used interchangeably.

*12) Line 86: Add arrows to Figure 1 (if keeping)* As suggested, we will remove Figure 1 (and section 2 entirely).

13) Line 89: For how long? Sense of time in Fig 1 is mostly missing. See replies #3 and #12.

*14) Line 90: Dune location needed in Fig 1a* See replies #3 and #12.

15) Line 92: Then why is this field study here? Not even one example run could be done to connect these observations with the SLF results? See replies #3 and #12.

16) Line 121: Seepage vector needs to be defined and developed more clearly. Not the same terminology as used later in either the methods or results. Assuming seepage means "specific discharge vector"? Why the magnitude? Isn't direction important? Figure 2 could be made more consistent with the methods/equations/variables. Where is this seepage vector being calculated (especially for Fig 2)? At the top of the free surface? At some depth? Figure 2 also does not show the "magnitude of seepage component" but the actual seepage component with changing directions. We will revise the sentence as follows:

The magnitude of the hydraulic head gradient (Figure 2), which according to Darcy's law is the magnitude of the seepage vector divided by the hydraulic conductivity, is denoted i (Figure 2). The seepage vector is the specific discharge, which is computed as the outflow vector at top nodes of the domain. In 2D, this vector has two components – a horizontal (-Kix in Figure 2) and a vertical (-Kiz). This work focuses on the vertical component. Other variables used in the following calculations are shown in Figure 2 and summarized in Table 1.

As suggested, we will also change Figure 2 for consistency with the text:

17) Line 144: Quick sand and liquefaction are related but exceptionally different. This is a weak justification. This analysis does not study liquefaction.

We will change all liquefaction to quicksand. Specifically, that sentence will be revised: Some publications distinguish between the Two terms that are often confused are "liquefaction" and "quick sand", with the former being used for earthquake-induced fluidization of the soil, and the latter being related to failure due to upward flow (Briaud, 2013). However, the The physical meaning of the two is the same similar – geomaterial becoming weightlesssuspended in a colloidal solution, which can result in erosion and sediment mobilization, or loss of support of any infrastructure built into the soil. Here, the term liquefaction-quicksand is used, although-as the analysis refers to surge-induced changes in the subsurface flow rather than seismically induced flows.

**18) Line 146: Sand is not weightless ever. It has mass. "Suspended" and liquid-like or colloidal-like, sure. Inaccurate terminology.**

See comment #17 – this entire sentence will be revised and instead of *weightless* we will use *suspended in a colloidal solution*

*19) Line 152: Full saturated or flooded?* We will change *inundated* to *flooded* as suggested.

20) Line 161: What simulated area? In the generic models. Suggests a field site. We will change to simulated topography.

21) Line 163: The use of however here implies there is a justification preceding it, but there is not. What is the "however" referring to? What is or is not being done in the study? A new paragraph may be helpful or a rearrangement of the two sentences that establish the differences.

This is related to comment #4 above, with better defining the differences between submerged and saturated unit weights. With the revision suggested there, we will change the following sentence accordingly, as follows:

The use of  $\gamma_{sub}$  as the representative unit weight of simulated soil is appropriate for soils that are fully submerged, as it accounts for the buoyancy effect, considering the unit weight of the overlying water column ( $\gamma_w$ ). However, for the parts of the model landward of the inundation line, the saturated unit weight may be more suitable. This means that adopting  $\gamma_{sub}$  uniformly may be an underestimate of the actual unit weight in real systems ( $\gamma_{sub} = \gamma_{sat} - \gamma_w$ ). However, Nevertheless, we used  $\gamma_{sub}$  since the aim of this work is to harness a hydrologic modeling framework...

**22) *Line 168: Enhance/increase/etc (alter in which way?)* We will change *alter* to *increase**

23) Line 178: Vz is the specific discharge according to this equation. Seepage velocity is easily confused with average linear velocity, and I do not see porosity in the equation. Providing units could also be useful to distinguish with the volumetric form.

Yes, it is the specific discharge, not the linear velocity (which is indeed why the porosity is absent from the equation. To make this clearer, we will change as follows:

According to Darcy's law the vertical flow velocitiesspecific discharge (denoted  $v_z$  with dimensions [MT]) are is equal to...

24) Line 185: And how are they different? A sentence is needed to explain the difference rather than the existence of a difference.

We will change the sentence to:

This result is similar to that derived by Briaud (2013) for a general case of quicksand. - but hHere it is derived specifically to facilitate for saturated groundwater flow, which is the appropriate formulation for the scenario of direct calculations of surge-induced changes in the groundwater flow regime as output by the hydrologic model.

25) Line 189: Positive and negative are confusing and dependent on the reference frame. When possible, it would help the reader to use upward and downward to describe flow. It would also be useful to minimize the use of gradient directions and stick with one convention that is clearly explained.

We agree that it is confusing and have tried to be consistent throughout the manuscript. Indeed, that sentence is particularly confusing since we explain there the use of absolute values for the gradient for convenience, since negative values are the potentially critical ones. To address this comment for that specific sentence we will add *upward* and *downward*, but prefer to leave *negative* and *positive* to explain the use of absolute values. The revised sentence will be:

For convenience, the magnitude of downward (negative (destabilizing) vertical head gradients which initiate upward (positive) vertical velocities and therefore potentially destabilize the soil, is hereinafter denoted  $i_z$  and presented in positive values.

**26) Line 191: Would be simple to provide a range of values**

Yes, certainly porosity and  $\rho_s$  have ranges rather than a specific values. However, we wanted to report the specific gradient value which we consider critical in this work, so we wrote the values plugged into Equation 7 to derive  $i_c = 0.15$ . To address this issue, we will change the following sentence to: Equation 7 suggests the critical value of vertical head gradient is about  $i_c = 0.15$ . While the parameters can have ranges of values for given systems, The the following analyses use this value as a threshold for liquefaction quicksand

27) Line 206: Please develop how this is a novel interdisciplinary approach within the context of the previous work (e.g., Yang and Tsai, 2020) and levee-based studies. This is the analysis of the outputs of a well-used groundwater model tool within a context that has been done before. I believe the study is worthwhile, but this statement is a gross and unjustified over-sell.

We agree that groundwater modeling is well used, although in this case we are using an *integrated hydrologic model* that couples surface and groundwater flows, which contributes somewhat to the novelty. We also agree that the context in which the model is used has been well studied. However, the combination of these two – harnessing an integrated hydrologic model in this context – is the novelty. Nonetheless, we agree that it is necessary to mention the recent work by Yang and Tsai (2020) and the similarities/differences, and will soften this statement as follows:

Hydrogeosphere has been successfully employed to simulate storm surges in several recent studies (Guimond & Michael, 2020; Yang et al., 2013, 2018; Yu et al., 2016), and here it is applied to assess the risk for sediment liquefaction quicksand and erosion from surge-induced pore water head gradients. This is a novel interdisciplinary approach, applying a robust 3D hydrologic model in the context of coastal geomechanics. This interdisciplinary approach, using a groundwater model in the context of coastal geomechanics, has recently been applied by Yang and Tsai (20

---

## Author Comment (AC2)

We have received the second referee's comments (RC2) on our paper, titled: **Hydrogeological controls on the spatio-temporal variability of surge-induced hydraulic gradients along coastlines: implications for beach surface stability** (A. Paldor et al.). We thank the referee for their valuable feedback, which we believe will improve the revised paper. Below is a list of the referee's comments (in red) and our respective replies (in black). Text cited from the manuscript is in black italicized, with suggested revisions highlighted in blue with tracked changes.

**General comments:**

**1) Line 32: I do not interpret the beach groundwater observations of Sous et al. 2016 as soil failure, please check.**

Agreed. In the revision we will leave only Stegmann et al. (2011) as a reference for that statement.

**2) Line 34: I do not fully agree with the definition given for liquefaction. A zero-stress soil needs an external force to be liquified.**

We agree, and will revise all *liquefaction* in the MS to *quicksand*, which, as the reviewer correctly points out, is the more suitable term in this context.

**3) *l.42: depends**

We agree and will correct to plural rather than changing the verb: Laboratory experiments (Sous et al., 2013) suggest that the magnitudes of hydraulic gradients in the beach due to infiltration from sea-swell and infragravity waves depend on the wave frequency, crossshore position, water table overheight, and the presence of standing waves.

**4) l. 53: Mory et al. 2007, I would emphasize here that liquefaction events were related to the presence of a rigid structure in the soil, and rather use Michallet et al. 2009 (JGR) for the same site but finer analysis.**

To address this comment, we will edit this sentence and add reference to the suggested study: *Observations, theories, and simulations have shown that the pore-pressure changes owing to energetic ocean waves can reduce effective stresses and may cause liquefaction\_failure of structures and surfaces (Chini & Stansby, 2012; Mory et al., 2007; Sakai et al., 1992; Sous et al., 2013; Yeh & Mason, 2014 Michallet et al., 2009*).

**5) *l.121: What is meant by "seepage vector"*?**

We will revise the sentence according to this comment:

The magnitude of the hydraulic head gradient (Figure 2), which according to Darcy's law is the magnitude of the seepage vector divided by the hydraulic conductivity, is denoted i (Figure 2). The seepage vector is the specific discharge, which is computed as the outflow vector at top nodes of the domain. In 2D, this vector has two components – a horizontal (-Kix in Figure 2) and a vertical (-Kiz). This work focuses on the vertical component. Other variables used in the following calculations are shown in Figure 2 and summarized in Table 1.

6) I do not see the input provided by Section 2. The data analysis has been already published, and the results presented here do not bring real insight (no vertical gradient, nothing new than much older works) and certainly do not show the statement in the Conclusions section ('may substantially affect...') We agree. In the revision we plan to remove Section 2 entirely (and Figure 1).

**7) l.163: Please detail the definition of unit weights and more generally provide a unified and clear discussion about saturation vs submersion effets (e.g. l. 292).**

To address this comment, we will edit in the explanation given for these two quantities (after equation 3): *The use of*  $\gamma$ *\_sub as the representative unit weight of simulated soil is appropriate for soils that are fully submerged, as it accounts for the buoyancy effect, considering the unit weight of the overlying water column (\gamma\_w*). However, for the parts of the model landward of the inundation line, the saturated unit weight may be more suitable. This means that adopting  $\gamma$ *\_sub uniformly may be an underestimate of the actual unit weight in real systems (\gamma\_sub* =  $\gamma$ *\_sat*- $\gamma$ *\_w*) We will also add these to Table 1.

**8) Can you justify the anistropy in K?**

To address this comment and the following one, we will edit the text in lines 222-223: The homogenous hydraulic conductivity Kx is 50 m/d for the baseline simulation, and values of Kx = 10, 25, 100 m/d were also simulated as part of a sensitivity analysis and Kx varied between 10 and 100 m/d in sensitivity analyses. In all simulations, the anisotropy was 10 (i.e., the vertical hydraulic conductivity, Kz, was 10 times lower than the horizontal hydraulic conductivity, Kx). This range of hydraulic conductivity with a porosity, n, of 0.3 is typical for sandy beach environments (Freeze and Cherry, 1979). Anisotropy of porous material may represent the presence of horizontally-extended low-K lenses (e.g., localized compacted clay lenses), which reduce the conductivity in the vertical dimension preferentially.

**9) Can you describe in detail your sensitivity analysis (parameters and ranges)?**

See previous comment for the suggested revision to address this. Additionally, we state these in lines 248-250:

The sensitivity of the results to the topography and hydrogeologic parameters was tested, including freshwater influx ( $0.01 < q_0 < 0.04 \text{ m/d}$ , Figure 3 and Table 1) and hydraulic conductivity (10 < Kx < 100 m/d, Table 1, typical values for sandy beaches (Freeze & Cherry, 1979)).

**10) The surge imposed here shows the same typical height and time scales than typical macrotidal areas. Does it mean that the potential "liquefaction" predicted here can be observed in any comparable macro-tidal coast ? Please comment.**

No, because in macro-tidal areas the dynamic steady state is different since the frequency of the fluctuations is still diurnal. Surges that occur over decadal time scales may induce quicks and as the sediment relaxation time scales is smaller. We will add this comment where the simulated surge height is reported:

For the transient surge simulations, the coastline head was varied over 8.5 hours between zero and a 3 m maximum surge height (inset in Figure 3). A sea level of 3 m above the mean represents a combined hightide and surge event with a projected return period of 100 yr by the year 2050 in the East Coast of the United States (Tebaldi et al., 2012). The ocean surface was assumed to be spatially constant at any time, and effects of wind waves were not simulated. The simulated surge height is comparable in magnitude to macro-tides, but the differences in frequency (macro-tides are diurnal) mean that macro-tidal beaches are likely in equilibrium with respect to sediment mobility, which is not the case for storm surges.

**11) - 1.308, 368 etc: What is meant by "overpressure"?**

We mean the pressure induced by the inundation water which is higher than the pre-surge pressure (steady state). To address this, we will change all *overpressures* in the manuscript to *increased pressures*.

**12) The role played by horizontal gradients is not explored, and this may significantly affect the interpretation.**

We agree that it is important to discuss horizontal gradients and we will revise section 6.1 as follows: Thus, the simulations suggest the areas most susceptible to destabilization (i.e., deep critical layers) are those where topography is low enough to be inundated widely, and high enough that the pressure release is limited. An important factor that likely plays a role in this relationship between intermediate topography and critical gradients is the horizontal gradient. In places where horizontal hydraulic gradients can develop, a more efficient dissipation of surge-induced pressures may be expected, and therefore critical gradients are less likely. This may explain the absence of critical hydraulic gradients from the steepest areas in the model, since these areas develop horizontal gradients. Horizontal gradients are important also when considering other modes of surface instability, such as shear failure. To assess the potential for shear failure, a Coulomb criterion must be derived, which is beyond the scope of the current study.

---

## Author Response (AR1)

We have received the reviews of two referees on our paper, titled: **Hydrogeological controls on the spatio-temporal variability of surge-induced hydraulic gradients along coastlines: implications for beach surface stability** (A. Paldor et al.). We thank the referees for their valuable feedback, which we believe improved the paper. Below is a list of the comments (in red) and our respective replies (in black). Text cited from the manuscript is in black italicized, with revisions highlighted in blue with tracked changes. All references to the text are to the revised manuscript with tracked-changes.

Referee #1 (RC1)
General comments:

*1) Quick sand is not the same thing as liquefaction. Quick sand describes a state of the material when it has no effective stress acting upon it. Liquefaction is the process of liquefying such a material with an outside force acting on it. Static liquefaction is possible (Sadrekarimi, 2014), but it also requires an outside force and describes a process. I do not believe that the term liquefaction is necessary for this study to be useful. Using incorrect and inaccurate terminology as the foundation is not useful. Quick sand more accurately describes the state being modeled, but perhaps that label is unwanted. No label is needed once the state is described. It may be most accurately termed a "colloidal solution". The state could be connected to the many potential hazards it can respond to/cause with external factors considered, including liquefaction, piping, and uplift. Groundwater modeling has been performed to understand these effects using the "Factor of Safety" ratio, which is 1/SLF as defined in this study for coastal settings (e.g., Yang & Tsai, 2020) as well as numerous applications to levees (as already referenced in the manuscript). Liquefaction potential related to sea-level rise causing higher water tables (i.e., changes in saturation) require specific earthquake events (e.g., Grant et al., 2021 and references therein). Including vertical head gradients could provide additional information to these analyses, but none of this was developed in the current manuscript. (Aside: A commentary on how sea-level rise could lead to more areas at risk of high head gradients with surges could be added to the implications discussion section).*

We agree with the reviewer that quicksand may be a more suitable term. We changed all instances of *liquefaction* to *quicksand* as the reviewer suggests. We also added text to clarify this in the introduction (lines 37-40):

*Soil liquefaction and quicksand occurs when pore pressures in the geomaterial rise to a point where its effective stress drops to zero and the material is fluidized, and thus acts as a liquid. The distinction between the two terms relates to the mechanism inducing the rise in pore pressures, with liquefaction referring to*

*cases where external forces (e.g., earthquakes) are involved. Quicksand is used for cases where the pore*

*pressures rise due to intrinsic changes in the groundwater regime.*

In section 2.1 in the revised manuscript, we also changed the title and text (lines 154-160):

***2.1 The criterion for  quicksand under groundwater seepage***

* Two terms that are often confused are "liquefaction" and
" quicksand", with the former being used for earthquake-induced fluidization of the soil, and
the latter being related to failure due to upward flow (Briaud, 2013).  The physical meaning
of the two is  similar – geomaterial becoming  suspended in a colloidal solution, which
can result in erosion and sediment mobilization, or loss of support of any infrastructure built into the soil.
Here, the term  quicksand is used,  as the analysis refers to surge-induced changes in
the subsurface flow rather than seismically induced flows.*

As for the reference to Yang and Tsai (2020), this is a very useful paper and we thank the reviewer for drawing our attention to it. We included in the revision references to this study in the Introduction (lines 69-71):

*Apart from waves, storm surges also could alter the onshore hydrogeological regime and potentially reduce
the stability of the beach surface. Recently, Yang & Tsai
(2020) modelled groundwater response to coastal flooding in the New Orleans greater area, and found that
the interaction between flood water and surface water may destabilize levees in the area.*

In the Methods we also added reference to this study when deriving the SLF, which as the reviewer correctly notices, is 1/FS as defined there (lines 214-219):

*In Equation 8, $i_z$ is the actual simulated or observed vertical head gradient, defined as $i_z = -\frac{v_z}{K_z}$ (Eq. 4)
and $i_c$ is the theoretical  quicksand threshold (Eq. 7). Thus, any point in space and time in
which simulated SLF is close to 1 is potentially nearing  quicksand. A layer in which SLF
approaches 1 continuously from the surface to a depth $Z_l$ is considered a "critical layer" of thickness $Z_l$.
The SLF defined here is the reciprocal of the Factor of Safety defined by Yang and Tsai (2020) for levees
under storm-induced groundwater seepage, and thus it should be noted that in the analysis presented
here lower values of SLF represent greater stability.*

We also discuss our modeling results in section 5.3 in comparison with the results Yang and Tsai (2020) obtained for the Greater New Orleans area (lines 473-475):

*While our hydrologic model is generalized, a recent study has showed that numerical hydrologic modelling
can be used to predict geomechanical risks induced by storm surges in specific settings too (Yang and Tsai,
2020).*

*2) Overpressures and excess pressures are not used in a context that I am familiar. This generally
implies greater than hydrostatic pressure (i.e., a confined or artesian pressure). I do not believe this is
what is intended. Rather, I interpret these terms to mean raised water levels from the surge event that*

*then need to drain away. The text does not make this clear, but it affects one of the main conclusions on the "intermediate" elevations playing an important role in setting low effective stress conditions.*

We agree, using the term over/excess pressure may be confusing with artesian conditions which are not considered here. We indeed mean groundwater pressures that are enhanced compared to calm conditions (i.e., surge-induced higher pressures). In the revision we changed all *over/excess pressures* to *increased pressures*.

*3) I do not find the "Field Evidence" portion of the manuscript to contribute to the overall purpose of the study. It appears to be based on previously published results (Housego et al., 2018). Referencing this study and moving on within the introduction to set the hydrogeologic context seems sufficient. If the model were more directly connected to the field site, then retaining such information could be useful. The conclusion that storm surge head gradients at the field site "substantially affect the stability of beach surfaces" is conjecture and no data are presented on beach stability or vertical hydraulic gradients (at least based on the information provided in Figure 1b). The response of the water table to a storm surge does not provide information on vertical gradients.*

We agree. In the revision we removed the field observations (Section 2 and Figure 1) from the manuscript.

*4) The difference between the submerged and saturated unit weights are not sufficiently clear to make them distinguishable other than that they are somehow different. Part of the problem is that the γsat and γfw are never defined. It is unclear how γsub would be different from γsat – the water level or pressure is not part of the equation, so it should not matter if the land surface is submerged or saturated. What is different about these two terms other than something to do with freshwater? Is γfw = 1? All of these unit weights need to be defined more clearly. I assume that these values depend on the modeled porewater salinity?*

To address this comment, we edited the explanation given for these two quantities (lines 173-179):

*The use of $\gamma_{sub}$ as the representative unit weight of simulated soil is appropriate for soils that are fully submerged, as it accounts for the buoyancy effect, considering the unit weight of the overlying water column ($\gamma_w$). However, for the parts of the model landward of the inundation line, the saturated unit weight may be more suitable. This means that adopting $\gamma_{sub}$ uniformly may be an underestimate of the actual unit weight in real systems ($\gamma_{sub} = \gamma_{sat} - \gamma_w$)*

We also added these to Table 1 (line 153).

*5) Varying the hydraulic conductivity a few times and considering two synthetic topographies represents a meager exploration of the parameter space. Interestingly, the parameters listed in Table 1 do not match the values used in the results figures and no justification for the ranges are provided (e.g., K in the 10-100 m/day range from Freeze and Cherry (1979) with no indication of sediment type and then a few values of Kz are reported in the later figures). How many models were performed? If only a few for each, then listing the exact values of the parameters seems feasible and useful. Using only one value of porosity for the critical index value seems overly simple. At a minimum, it could be useful to present what the range of the SLF index is for realistic coastal geomaterials.*

We agree that further exploration of the parameter space would be useful and interesting. However, this study is aimed to show the importance alongshore topographic variability and to propose an approach for better tying hydrogeologic and geomechanical modeling/measurements. To that end, a more detailed exploration of the parameter space may be off point. In the revision, we stressed this valuable point at the end of section 3 with the following text (lines 298-303):

*It is noted that exploring 4 values of hydraulic conductivity and two types of synthetic topographies may be a limited representation of natural systems. For example, Xu et al. (2016) showed that topographic connectivity is a dominant factor in the vulnerability of coastal aquifers to storm surge salinization, and we consider here only two of the topographies simulated there. However, the tested topographies and conductivities in this work serve as a preliminary exploration of hypothetical conditions that are likely representative of many natural systems, but is certainly not inclusive.*

As for the hydraulic conductivities, we state the values and their suitability for beach sediments, and also the Kz values. Additionally, Figure A4 in the appendices details the values simulated. To address this comment and to better clarify, we revised as follows (lines 252-256):

*The homogenous hydraulic conductivity Kx is 50 m/d for the baseline simulation, and values of Kx = 10, 25, 100 m/d were also simulated as part of a sensitivity analysis. In all simulations, the anisotropy was 10 (i.e., the vertical hydraulic conductivity, Kz, was 10 times lower than the horizontal hydraulic conductivity, Kx). This range of hydraulic conductivity with a porosity, n, of 0.3 is typical for sandy beach environments (Freeze and Cherry, 1979).*

*6) Conceptually, I feel like the main conclusion that "topography" is important for setting the colloidal solution development is not fully explored. No alternative hypotheses or more nuanced hydrologic explanations are tested, even though the modeling produces the outputs to test them. The elevation of the land surface and ability to drain to the coast are invoked as the primary driver of the "intermediate" elevation cases where the lingering higher pressures occur. This is reasonable to me, but I believe the mechanism is more related to two controls 1) the initial water table depth setting the infiltration capacity and rate(s) as seen with tsunamis and storm surges (Cardenas et al., 2015), and 2) the horizontal vs vertical gradients allowing the pressure to dissipate in those areas. Despite the relative complexity of the synthetic topographies, it should be possible to normalize model inputs and outputs and explore these controls more deeply.*

We agree that it is important to acknowledge these potential controls and we added the following text to section 5.1 (lines 424-433):

*Thus, the simulations suggest the areas most susceptible to destabilization (i.e., deep critical layers) are those where topography is low enough to be inundated widely, and high enough that the pressure release is limited. An important factor that likely plays a role in this relationship between intermediate topography and critical gradients is the horizontal gradient. In places where horizontal hydraulic gradients can develop, a more efficient dissipation of surge-induced pressures may be expected, and therefore critical gradients are less likely. This may explain the absence of critical hydraulic gradients from steepest areas in the model, since these areas develop horizontal gradients. Horizontal gradients are important also when considering other modes of surface instability, such as shear failure. To assess the potential for shear failure, a Coulomb criterion must be derived, which is beyond the scope of the current study. Another factor that is known to control the vulnerability to storm-induced instability is the antecedent groundwater level which controls the infiltration capacity of flood waters (Cardenas et al., 2015). This may explain the absence of critical hydraulic gradients from the flatter areas of the model, leaving an intermediate range of topographies that are susceptible to surge-induced critical gradients.*

Specific comments:

*7) I do not find that the title explains the study well. "Topographic controls on surge-induced critical groundwater gradients and potential surface instability"*

We agree that the title should mention the topographic controls. We revised to (lines 1-5):

*Coastal topography and hydrogeology control critical groundwater gradients and potential beach surface instability during storm surges*

*8) Line 24: Revised to ... - explain what the revision should be "to include steeper portions of ..."*
Agreed, changed accordingly (lines 25-27):
*These findings suggest that the common practices in monitoring and mitigating surge-induced failures and erosion, which typically focus on the flattest areas of beaches, might need to be revised to include other topographic features.*

*9) Line 26: Typo*
We changed *especFially* to *especially* (line 29).

*10) Line 37: Pressure distributions do not induce surface failure. An outside force/stress is required to induce this.*
We revised that sentence (lines 40-43):
*At the coast, ocean (waves, surge, tides, inundation) and terrestrial (groundwater heads, precipitation, and overland flows) processes concurrently contribute to changing pore pressures in beach and nearshore sediments, and*  *could thus induce failure of the surface.*

*11) Line 66: Flooding is generally a temporary event while inundation implies a longer-term water coverage. I believe all uses of "inundation" in this manuscript refer to flooding.*
We changed all inundation to flooding, though the terms are often used interchangeably.

*12) Line 86: Add arrows to Figure 1 (if keeping)*
As suggested, we removed Figure 1 (and section 2 entirely).

*13) Line 89: For how long? Sense of time in Fig 1 is mostly missing.*
See replies #3 and #12.

*14) Line 90: Dune location needed in Fig 1a*
See replies #3 and #12.

*15) Line 92: Then why is this field study here? Not even one example run could be done to connect these observations with the SLF results?*
See replies #3 and #12.

*16) Line 121: Seepage vector needs to be defined and developed more clearly. Not the same terminology as used later in either the methods or results. Assuming seepage means "specific discharge vector"? Why the magnitude? Isn't direction important? Figure 2 could be made more consistent with the methods/equations/variables. Where is this seepage vector being calculated (especially for Fig 2)? At the top of the free surface? At some depth? Figure 2 also does not show the "magnitude of seepage component" but the actual seepage component with changing directions.*
We revised the sentence as follows (lines 130-134):
*The magnitude of the hydraulic head gradient* *, which according to Darcy's law is the magnitude of the seepage vector divided by the hydraulic conductivity, is denoted i (Figure 1). The seepage vector is the specific discharge, which is computed as the outflow vector at top nodes of the*

*domain. In 2D, this vector has two components – a horizontal (-Ki_x in Figure 2) and a vertical (-Ki_z). This work focuses on the vertical component.* Other variables used in the following calculations are shown in Figure 2 and summarized in Table 1.

As suggested, we also changed Figure 2 (which is now Figure 1) for consistency with the text:

[Figure]

[Figure]

*17) Line 144: Quick sand and liquefaction are related but exceptionally different. This is a weak justification. This analysis does not study liquefaction.*
We changed all *liquefaction* to *quicksand*. Specifically, that text is now revised (lines 155-160):
 Two terms *that are often confused are* "liquefaction" and "quicksand", with the former being used for earthquake-induced fluidization of the soil, and the latter being related to failure due to upward flow (Briaud, 2013).  The physical meaning of the two is  similar – geomaterial becoming suspended in a colloidal solution, which can result in erosion and sediment mobilization, or loss of support of any infrastructure built into the soil. Here, the term  quicksand is used,  as the analysis refers to surge-induced changes in the subsurface flow rather than seismically induced flows.

*18) Line 146: Sand is not weightless ever. It has mass. "Suspended" and liquid-like or colloidal-like, sure. Inaccurate terminology.*
See comment #17 – this entire sentence has been revised and instead of *weightless* we use *suspended in a colloidal solution* (line 158).

*19) Line 152: Full saturated or flooded?*
We changed to *flooded* as suggested.

*20) Line 161: What simulated area? In the generic models. Suggests a field site.*

We changed to *simulated topography* (line 173).

*21) Line 163: The use of however here implies there is a justification preceding it, but there is not. What is the "however" referring to? What is or is not being done in the study? A new paragraph may be helpful or a rearrangement of the two sentences that establish the differences.*

This is related to comment #4 above, with better defining the differences between submerged and saturated unit weights. With the revision there, we have changed the following sentence accordingly, as follows (lines 179-181):

*Nevertheless, we used $\gamma_{sub}$since the aim of this work is to harness a hydrologic modelling framework to assess the spatio-temporal distribution of surge-induced changes in hydraulic gradients.*

*22) Line 168: Enhance/increase/etc (alter in which way?)*

We changed *alter* to *increase* (line 184)

*23) Line 178: Vz is the specific discharge according to this equation. Seepage velocity is easily confused with average linear velocity, and I do not see porosity in the equation. Providing units could also be useful to distinguish with the volumetric form.*

Yes, it is the specific discharge, not the linear velocity (which is indeed why the porosity is absent from the equation. To make this clearer, we changed as follows (lines 192-193):

*According to Darcy's law the vertical specific discharge (denoted $v_z$ with dimensions [MT$^{-1}$])) is equal to*

*24) Line 185: And how are they different? A sentence is needed to explain the difference rather than the existence of a difference.*

We changed the sentence to (lines 201-203:

*This result is similar to that derived by Briaud (2013) for a general case of quicksand. Here it is derived specifically to facilitate  direct calculations of surge-induced changes in the groundwater flow regime as output by the hydrologic model.*

*25) Line 189: Positive and negative are confusing and dependent on the reference frame. When possible, it would help the reader to use upward and downward to describe flow. It would also be useful to minimize the use of gradient directions and stick with one convention that is clearly explained.*

We agree that it is confusing and have tried to be consistent throughout the manuscript. Indeed, that sentence is particularly confusing since we explain there the use of absolute values for the gradient for convenience, since negative values are the potentially critical ones. To address this comment for that specific sentence we added *upward* and *downward*, but prefer to leave *negative* and *positive* to explain the use of absolute values. The revised sentence is (lines 206-208):

*For convenience, the magnitude of downward (negative ) vertical head gradients which initiate upward (positive) vertical velocities and therefore potentially destabilize the soil, is hereinafter denoted $i_z$ and presented in positive values.*

*26) Line 191: Would be simple to provide a range of values*

Yes, certainly porosity and $\rho_s$ have ranges rather than specific values. However, we wanted to report the specific gradient value which we consider critical in this work, so we wrote the values plugged into Equation 7 to derive $i_c = 0.15$. To address this issue, we changed the following sentence to (lines 209-211):

*Equation 7 suggests the critical value of vertical head gradient is about $i_c = 0.15$. While the parameters can have ranges of values for given systems, the following analyses use this value as a threshold for quicksand*

*27) Line 206: Please develop how this is a novel interdisciplinary approach within the context of the previous work (e.g., Yang and Tsai, 2020) and levee-based studies. This is the analysis of the outputs of a well-used groundwater model tool within a context that has been done before. I believe the study is worthwhile, but this statement is a gross and unjustified over-sell.*

We agree that groundwater modeling is well used, although in this case we are using an *integrated hydrologic model* that couples surface and groundwater flows, which contributes somewhat to the novelty. We also agree that the context in which the model is used has been well studied. However, the combination of these two – harnessing an integrated hydrologic model in this context – is the novelty. Nonetheless, we agree that it is necessary to mention the recent work by Yang and Tsai (2020) and the similarities/differences, and have softened this statement as follows (lines 228-236):

*Hydrogeosphere has been successfully employed to simulate storm surges in several recent studies (Guimond & Michael, 2020; Yang et al., 2013, 2018; Yu et al., 2016), and here it is applied to assess the risk for  quicksand and erosion from surge-induced pore water head gradients.  This interdisciplinary approach, using a groundwater model in the context of coastal geomechanics, has recently been applied by Yang and Tsai (2020) to assess the impacts of floods on the groundwater regime in the Greater New Orleans area, and its implications for the factor of safety of levees. Several other studies have also applied different methods to relate between changes in the groundwater regime and the stability of the surface (Chini & Stansby, 2012; Sakai et al., 1992; Sous et al., 2013; Yeh & Mason, 2014). The novelty in this study relates to the harnessing of a 3D integrated hydrologic model in a generalized form to explore the mechanisms that dominate surge-induced quicksand formation. Applying the fully-coupled model on different generalized topographies (detailed below) allows us to study the alongshore distribution of critical gradients, which is commonly overlooked in similar studies (Yeh and Mason, 2014).*

*28) Lines 214-219: Why are these results sentences in the methods? Seem out of place.*

These are not results that are meant to drive the points of the paper, but to justify the choice of modeled slopes and coastline locations, which is part of the methods. To clarify this, we revised as follows (lines 244-249):

*To justify this setting,  we ran a simulation with a -0.5 m sea level (i.e., still water shoreline at X=225 m), which  indicated that critical vertical hydraulic gradients occur near this change in overall slope irrespective of the shoreline location (Figure A1 in the Appendices). A simulation with a larger beach slope (Z(X=0)= -6;slope=6/450=0.0130) resulted in similar vertical hydraulic gradients as the baseline slope (0.0022) (Figure A2 in the Appendices), indicating that although the baseline slope is lower than typical, the analysis based on it is also valid for steeper slopes.*

*29) Line 258: Are crater topographies a standard coastal type? "Closed-dune" or similar would get the point across without invoking the extraterrestrial.*

Indeed, crater topographies may not be typical of coastal settings. However, simulating this topography serves the main purpose of this study – to better understand the importance of topography for the distribution of surge-induced critical hydraulic gradients. This is also noted in the text in the same place specified by the reviewer, and to address this comment we better emphasized (lines 291-298):

*The second topography, "Crater" (Figure 4b), features connected crests surrounding disconnected surface depressions, such that the highs are connected, forming "crater" like shapes. The two topographies do not mirror each other (Figure 4), but represent reverse alongshore trends near the shoreline (450<X<500 m) in which the area around 0<Y<300 m (2200<Y<2500 m) is the highest (lowest) for the River topography and lowest (highest) for the Crater topography. Comparisons with real topographies of the Delaware coastal plains (Yu et al. 2016) suggested that the River topography best represents real-world meso-topography. However, the Crater topography provides important insights to*

*how meso-topography controls the evolution of head gradients during storm surges even though they are not necessarily representative of real systems.*

*30) Line 275: Doesn't Figure 5 show 3D volumes of the SLF? The vertical slices appear to be a visualization technique rather than "the only place we have results".*

Agreed. We revised that sentence (lines 315-317)*:*
*For each simulation, the vertical hydraulic gradients (i_z in Equation 8) are calculated for the modeled domain  and normalized by the threshold defined by Equation 7 (i_c) to calculate the SLF (Equation 8).*

*31) Line 276: Provide the value of the threshold in this sentence and remove the next.*

We wanted to write this part as a standalone, so that the reader does not have to go back to section 3 to understand. Therefore, we broke it up into two sentences – one as a reminder of the definitions and another as a reminder of the threshold. Condensing all this into one sentence produced an overly complex sentence.

*32) Line 291: Unit weights difference still uninterpretable*

We have now better defined this following comment #4 above. We will also edit here and reference to the revised definition for more clarity (lines 330-334):
*The vertical hydraulic gradients onshore of the  flooding front during run-up (Figure 5b) develop in subaerial areas. As explained in section 3.1 above,  the calculated SLF for these zones  should be based on the saturated unit weight (γ_sat=γ_sub+γ_fw) of sediments rather than the submerged unit weight (γ_sub, Equation 3), and the model-predicted  quicksand may not occur in real systems because saturated soils are more stable than submerged ones (Briaud, 2013).*

*33) Line 308: Not overpressure.*

See the reply to comment #2 above. In the revision we changed all *over/excess* to *increased*.

*34) Line 309: Which head differences? In space or time? From what to what?*

We agree it is confusing and have edited this sentence (lines 350-352):
*The temporal differences in head  between surge and calm conditions also are low in the topographic highs because the heads there did not rise significantly during flooding.*

*35) Line 313: Not excess head*
We changed to *increased head*.

*36) Line 328: And why does this matter?*

To address this comment, we added the following text to that sentence (lines 370-372):
*However, in both cases this area is where the least significant vertical head gradients develop (Figure  5c1 and c2). This means that a monotonic relationship cannot be assumed between topography and vulnerability (i.e., the lowest/highest areas along the beach are not necessarily the most/least vulnerable).*

*37) Line 348: Justification is "because there's more of them"? Really? The nice idea with the Factor of Safety (1/SLF) is that you don't want to be exactly at the critical value but need some buffer. Explaining this choice within a FoS framework would provide stronger justification.*

This comment depends on the goal of the study. If the goal is to evaluate the geotechnical stability of a given area, then we agree that the important thing is the factor of safety. But, for this hypothetical study, having more results to better establish a correlation is the primary goal. To satisfy this comment, we added to that sentence what the reviewer suggests (lines 392-394):

*Here, the SLF=0.7 contour is used because for engineering applications it is required to design structures with a buffer to ensure a satisfactory factor of safety. Furthermore, using the SLF=0.7 provides better  statistical stability since there are more locations with SLF≥0.7 than with SLF=1.*

**38) Line 370: What prevents this release? Slow drainage? Long flow paths? Other? Developing this would allow the next statement on something explaining the similarities (correlation? statistical danger word)**

To address both of these valid comments, we edited that sentence (lines 415-418):

*Topographic elements that are low enough to be inundated, but are also high enough to limit the post-surge exfiltration may prevent release of pressures with thicker porous medium that impedes flow, possibly explaining the  link between quicksand potential  and intermediate topographic features (1-3 m high for a 3 m surge).*

**39) Line 376: Implications for real systems with more anisotropy and heterogeneity could be developed here with reference support on the importance of such things in coastal groundwater hydrodynamics.**

Agreed, we developed this as suggested with the following addition (lines 434-439):

*A simulation with even lower hydraulic conductivity (Kz=0.05) showed that very low values of K limit the surge-induced infiltration and thus critical gradients develop only to a limited vertical extent and the alongshore variability (i.e., the dependency on onshore topography) diminishes (Figure A5 in the Appendices). This result has important implications to systems with higher clay content, since lower K values may mean that beach topography controls the overall vulnerability less than in sandy beaches.*

**40) Line 383: Arguably, none of these results suggest liquefaction, but it does seem important to develop this further to explain why intruding flood waters with the same pressure gradients lead to different results.**

To address this issue, we revised the sentence there (lines 444-446):

*However, the simulated values of SLF=1 inland of the inundation front  do not necessarily imply  that quicksand is expected there in real systems, because the actual weight of the unsubmerged soil is greater than the uniformly-modeled $\gamma\_sub$ (Equation 2).*

**41) Line 386: How does this pressure (really pressure head) difference relate to those modeled in this study? Would help provide context of 0.01 m to an actual range – does this matter or not?**

We agree and revised accordingly (lines 446-451):

*Nevertheless, the  quicksand potential calculated here may still represent an underestimate, as Mory et al. (2007) showed that as little as 6% air content in the pores may reduce the pressure head required  liquefy the sediment by 0.01 m. While this 1 cm difference is an order of magnitude lower than the head changes discussed here (Figure 6), it is possible that in other hydrogeological settings the air content is more influential and therefore assuming fully saturated conditions may be a substantial underestimate of the quicksand potential.*

**42) Line 394: This result is not presented – the "simultaneously reached". Only snapshots of model results are show, and no time-dependent results are provided to support this portion of the discussion.**

This can be seen in Figure 5, where all the red patches in the shore-parallel panels appear at t=8.4 hr. Following the reviewer's comment, a reference to this figure was added to the text (lines 457-459):

*For all simulations at all alongshore locations, the positive head gradients simultaneously reached a maximum when the water had receded completely (t=8.4 hr, Figure 4d) and all the inundation water overburden was released.*

**43) Line 396: Indication that the flow rates related to infiltration rates are an important control and should be analyzed. This sentence also says that lowering pressure reduces the pressure gradient and is self-evident. Lowering a numerator does have that effect on a fraction.**

Thanks to this comment we noticed a confusion in that sentence and fixed it in the revision (lines 459-461):

*The rate of head release determines the hydraulic gradients that occur in the soil material, so that faster release of the  increased pressures allows less dissipation of elevated heads in the soil and therefore produces thicker critical layers.*

*44) Line 404: Maximum {vertical} hydraulic gradients. Directions needed.*
We agree and added *vertical* as suggested (line 469).

*45) Lines 421-422: This was in no way a result of this study.*
Following the comment about the connection between the field observations and the modeling (comment #3 above) we removed this from the Conclusions too. The opening paragraph of the conclusions is changed to (lines 486-492):

*Storm surges may substantially affect the groundwater regime in flooded areas, which can reduce the stability of beach surfaces. We explored this idea and its generality by harnessing a robust hydrological model to simulate a generalized coastal system and found that in the nearshore area, surge-induced hydraulic gradients may peak to critical levels that could potentially induce quicksand.*

*46) Figure 2: A hypothetical system. Need consistency with methods. Missing subscript.*
We edited the caption as suggested (lines 136-141):

*A  hypothetical coastal hydrogeological system. Regional fresh (light blue) groundwater flows to the sea and upward due to variable-density flow along the freshwater-saltwater (red) interface. In the nearshore area, focused groundwater discharge occurs either into the sea (blue) or along a seepage face onshore. As shown in the top of the figure, when the surge begins, the direction of flow reverses (infiltration), and when the sea level reaches its maximal level ($\text{}h_{max}$) the surge retreats and the direction reverts back (exfiltration). The upward (positive vertical component) of flow reaches a maximum when the sea level is back to pre-surge level, before decaying to the steady-state magnitude.*

*47) Figure 5: Are the 3D color volumes the vertical hydraulic gradient or SLF? The last sentence of the caption implies they are gradients, but there is no colorbar for gradient.*
We agree, the last sentence in the caption is confusing and to address this we edited as follows (lines 341-342):

*Note that downward gradients (head increases downward) are plotted as positive values of SLF and upward gradients (head increases upward) are plotted as zero SLF.*

*48) Figure 7: I suggest removing Figure 7 and incorporating these results into Figure 6, which would benefit from a contoured SLF value and topography included above the depth slices in c).*
We agree they could be merged, but prefer to leave them as separate figures because Figure 6 (now Figure 5) is overwhelming as it is.

General comments:

*1) Line 32: I do not interpret the beach groundwater observations of Sous et al. 2016 as soil failure, please check.*

Agreed. In the revision we left only Stegmann et al. (2011) as a reference for that statement (line 36).

*2) Line 34: I do not fully agree with the definition given for liquefaction. A zero-stress soil needs an external force to be liquified.*

We agree, and have revised all *liquefaction* in the MS to *quicksand*, which, as the reviewer correctly points out, is the more suitable term in this context.

*3) l.42: depends*

We agree and corrected to plural rather than changing the verb (lines 46-48):

*Laboratory experiments (Sous et al., 2013) suggest that the magnitudes of hydraulic gradients in the beach due to infiltration from sea-swell and infragravity waves depend on the wave frequency, cross-shore position, water table overheight, and the presence of standing waves.*

*4) l. 53: Mory et al. 2007, I would emphasize here that liquefaction events were related to the presence of a rigid structure in the soil, and rather use Michallet et al. 2009 (JGR) for the same site but finer analysis.*

To address this comment, we edited this sentence and added reference to the suggested study (lines 58-60):

*Observations, theories, and simulations have shown that the pore-pressure changes owing to energetic ocean waves can reduce effective stresses and may cause  failure of structures and surfaces (Chini & Stansby, 2012; Mory et al., 2007; Sakai et al., 1992; Sous et al., 2013; Yeh & Mason, 2014 Michallet et al., 2009).*

*5) l.121: What is meant by "seepage vector" ?*

We revised the sentence according to this comment (lines 130-134):

*The magnitude of the hydraulic head gradient , which according to Darcy's law is the magnitude of the seepage vector divided by the hydraulic conductivity, is denoted i (Figure 2). The seepage vector is the specific discharge, which is computed as the outflow vector at top nodes of the domain. In 2D, this vector has two components – a horizontal ($-Ki_x$ in Figure 1) and a vertical ($-Ki_z$). This work focuses on the vertical component. Other variables used in the following calculations are shown in Figure 2 and summarized in Table 1.*

*6) I do not see the input provided by Section 2. The data analysis has been already published, and the results presented here do not bring real insight (no vertical gradient, nothing new than much older works) and certainly do not show the statement in the Conclusions section ('may substantially affect…')*

We agree. In the revision we removed Section 2 entirely (and Figure 1).

*7) l.163: Please detail the definition of unit weights and more generally provide a unified and clear discussion about saturation vs submersion effets (e.g. l. 292).*

To address this comment, we edited in the explanation given for these two quantities (lines 175-179):

**

*The use of $\gamma_{sub}$ as the representative unit weight of simulated soil is appropriate for soils that are fully submerged, as it accounts for the buoyancy effect, considering the unit weight of the overlying water column ($\gamma_w$). However, for the parts of the model landward of the inundation line, the saturated unit weight may be more suitable. This means that adopting $\gamma_{sub}$ uniformly may be an underestimate of the actual unit weight in real systems ($\gamma_{sub} = \gamma_{sat} - \gamma_w$)*

We also added these to Table 1 (line 153).

*8) Can you justify the anistropy in K ?*

To address this comment and the following one, we edited the text in lines 252-258:

*The homogenous hydraulic conductivity Kx is 50 m/d for the baseline simulation, and values of Kx = 10, 25, 100 m/d were also simulated as part of a sensitivity analysis. In all simulations, the anisotropy was 10 (i.e., the vertical hydraulic conductivity, Kz, was 10 times lower than the horizontal hydraulic conductivity, Kx). This range of hydraulic conductivity with a porosity, n, of 0.3 is typical for sandy beach environments (Freeze and Cherry, 1979).Anisotropy of porous material may represent the presence of horizontally-extended low-K lenses (e.g., localized compacted clay lenses), which reduce the conductivity in the vertical dimension preferentially.*

*9) Can you describe in detail your sensitivity analysis (parameters and ranges) ?*

See previous comment for the revision we incorporated to address this. Additionally, we state these in lines 282-284:

*The sensitivity of the results to the topography and hydrogeologic parameters was tested, including freshwater influx (0.01< q_0 < 0.04 m/d, Figure 3 and Table 1) and hydraulic conductivity (10 < Kx < 100 m/d, Table 1, typical values for sandy beaches (Freeze & Cherry, 1979)).*

*10) The surge imposed here shows the same typical height and time scales than typical macrotidal areas. Does it mean that the potential "liquefaction" predicted here can be observed in any comparable macro-tidal coast ? Please comment.*

No, because in macro-tidal areas the dynamic steady state is different since the frequency of the fluctuations is still diurnal. Surges that occur over decadal time scales may induce quicksand as the sediment relaxation time scales is smaller. We added this comment where the simulated surge height is reported (lines 276-281):

*For the transient surge simulations, the coastline head was varied over 8.5 hours between zero and a 3 m maximum surge height (inset in Figure 3). A sea level of 3 m above the mean represents a combined high-tide and surge event with a projected return period of 100 yr by the year 2050 in the East Coast of the United States (Tebaldi et al., 2012). The ocean surface was assumed to be spatially constant at any time, and effects of wind waves were not simulated. The simulated surge height is comparable in magnitude to macro-tides, but the differences in frequency (macro-tides are diurnal) mean that macro-tidal beaches are likely in equilibrium with respect to sediment mobility, which is not the case for storm surges.*

*11) - l.308, 368 etc: What is meant by "overpressure" ?*

We mean the pressure induced by the inundation water which is higher than the pre-surge pressure (steady state). To address this, we changed all *overpressures* in the manuscript to *increased pressures*.

*12) The role played by horizontal gradients is not explored, and this may significantly affect the interpretation.*

We agree that it is important to discuss horizontal gradients and we revised section 6.1 as follows (lines 423-430):

*Thus, the simulations suggest the areas most susceptible to destabilization (i.e., deep critical layers) are those where topography is low enough to be inundated widely, and high enough that the pressure release is limited. An important factor that likely plays a role in this relationship between intermediate topography and critical gradients is the horizontal gradient. In places where horizontal hydraulic gradients can develop, a more efficient dissipation of surge-induced pressures may be expected, and therefore critical gradients are less likely. This may explain the absence of critical hydraulic gradients from the steepest areas in the model, since these areas develop horizontal gradients. Horizontal gradients are important also when considering other modes of surface instability, such as shear failure. To assess the potential for shear failure, a Coulomb criterion must be derived, which is beyond the scope of the current study.*